# SEMI-ANCHORED GRADIENT METHODS FOR NONCONVEX-NONCONCAVE MINIMAX PROBLEMS

## ABSTRACT

Nonconvex-nonconcave minimax problems are difficult to optimize by gradient methods. The extragradient method, proven to outperform the gradient descent ascent, has become standard but there is still room for improvement in terms of a convergence speed. On the other hand, under a bilinear setting, the primal-dual hybrid gradient (PDHG) method is one of the most popular methods, and is superior to the extragradient in practice. The PDHG was studied on a convex-concave problem, but it has not been found useful in a more general nonconvex-nonconcave minimax problem. In this paper, we demonstrate a natural extension of the PDHG to a structured nonconvex-nonconcave minimax problem, whose saddle-subdifferential operator satisfies the weak Minty variational inequality condition. This new nonlinear variant of PDHG, named semi-anchored (SA) gradient method, is built upon the Bregman proximal point method. This consequently provides a worst-case convergence rate, in terms of a new Bregman distance-based optimality measure. This rate analysis of the SA implies a possible improvement over the extragradient with a similar rate in terms of the squared gradient norm, as the new optimality measure upper bounds the squared gradient norm. In addition, since the SA exactly reduces to the PDHG for a bilinear problem, it is likely to be faster than the extragradient in practice when entered a locally bilinear region. We further illustrate the potential of the SA, by providing a fair classification experiment, where it outpaces the extragradient, given an efficient max-oracle.

## 1 INTRODUCTION

Generative adversarial network (GAN) (Arjovsky et al., 2017; Goodfellow et al., 2014), adversarial training (Kurakin et al., 2017; Mądry et al., 2018) and fair training (Mohri et al., 2019; Nouiehed et al., 2019) involve solving a nonconvex-nonconcave minimax problem. These applications have produced numerous promising results, but their gradient-based training is yet reported to be difficult and laborious. In minimax optimization, the extragradient method (Korpelevich, 1976) has become one of standard methods in minimax optimization, as it is known to outperform gradient descent ascent method (Mescheder et al., 2018; Gorbunov et al., 2022). However, there is still room for improvement, in terms of a convergence speed, possibly from a different perspective.

For a (composite) bilinear problem, the primal-dual hybrid gradient (PDHG) method (Chambolle & Pock, 2011; Esser et al., 2010) is one of the favorites, and practically outperforms the extragradient method (Chambolle & Pock, 2011). This was extended to convex-concave problems (Hamedani & Aybat, 2021; Yadav et al., 2018; Zhao, 2019), but has not been studied in a more general nonlinear nonconvex-nonconcave minimax problems. This paper thus constructs a new nonlinear variant of the PDHG, named semi-anchored (SA) gradient method, that shows several theoretical and empirical improvements over existing methods.

The proposed SA gradient method is built upon the theory of the Bregman proximal point (BPP) method (Bauschke et al., 2003; Borwein et al., 2011; Eckstein, 1993). In specific, the PDHG method is an instance of the proximal point method (Martinet, 1970; Rockafellar, 1976) with a specific preconditioner (He & Yuan, 2012). The BPP method is a nonlinear extension of the proximal point method via the Bregman distance (Bregman, 1967), and we choose one that directly extends the preconditioner of the PDHG. We consequently show that the SA gradient method finds a stationary point of a structured nonconvex-nonconcave composite problem; its Lipschitz continuous

saddle-subdifferential operator satisfies the weak Minty variational inequality (MVI) condition in (Diakonikolas et al., 2021). The weak MVI condition is weaker than the MVI condition that has received recent attention as one of standard nonconvex-nonconcave settings (Dang & Lan, 2015; Malitsky, 2020; Mertikopoulos et al., 2019; Song et al., 2020; Zhou et al., 2017).

For the worst-case rate analysis, we first show that the worst-case rate of the BPP method is $O(1/k)$, under the weak MVI condition, in terms of the Bregman distance between the successive iterates, where $k$ denotes the number of iterations. Then, the *exact* SA method, named SA-GDmax, using an exact maximization oracle for the max-player, consequently has the same $O(1/k)$ rate. Here, the considered optimality measure, tailored from the Bregram distance, which is new in minimax optimization literature. The extragradient method also has the $O(1/k)$ rate (Diakonikolas et al., 2021; Gorbunov et al., 2022) but in terms of the squared gradient norm that is upper bounded by our new optimality measure. This implies that the SA-GDmax can be superior to the extragradient in the worst case. Also note that the SA-GDmax exactly reduces to the PDHG for a bilinear problem, so we can expect that it will practically outperform the extragradient when entered a locally bilinear region. We then show that its more *practical* version, named SA-MGDA, using a finite number of gradient steps of the max-player, has a complexity that matches that of the *exact* version up to a log factor.

Our theoretical results of the SA gradient method, built upon the PDHG and the BPP, are comparable to those of extragradients (as detailed later), showing potential of the PDHG-like methods. We further provide numerical experiments where the proposed SA-GDmax outperforms extragradient, given an efficient max-oracle, making the PDHG-like methods more interesting in general minimax problems.

**Our main contributions** are summarized as follows.

- We study the properties of the BPP method and analyze its worst-case rate, in terms of the Bregman distance between two successive iterates, under the weak MVI condition, in Section 4. We also similarly study a version of the BPP with projection onto a separating hyperplane that works for a larger range of the weak MVI condition.
- Built upon Section 4, we develop a new semi-anchoring (SA) approach, a new nonlinear variant of the PDHG in Section 5. We further construct its inexact but more practical variant, named SA-MGDA.
- In Section 6, we present the worst-case rates of the SA gradient methods for structured nonconvex-nonconcave composite problems. This analysis is based on a newly introduced Bregman distance-based optimality measure, and the worst-case rate is comparable to (and possibly better than) the extragradient method. We also analyze the SA gradient methods under the strong MVI condition (Song et al., 2020; Zhou et al., 2017), in Appendix E.
- Section 7 provides two numerical experiments, where an efficient maximization oracle is available and consequently the SA gradient method outperforms the extragradient.

## 2 PRELIMINARIES

### 2.1 BREGMAN DISTANCE

This paper uses a Legendre function (Rockafellar, 1970) and its associated Bregman distance (Bregman, 1967), defined below, as a non-Euclidean proximity measure (Eckstein, 1993; Teboulle, 2018). These help us to better handle the nonlinear geometry of a problem.

**Definition 1.** *Let $h : \mathbb{R}^d \to (-\infty, \infty]$ be a Legendre function (Rockafellar, 1970). The Bregman distance associated to $h$, denoted by $D_h : \operatorname{dom} h \times \operatorname{int} \operatorname{dom} h \to \mathbb{R}_+$ is defined by $D_h(\boldsymbol{x}, \boldsymbol{y}) := h(\boldsymbol{x}) - h(\boldsymbol{y}) - \langle \nabla h(\boldsymbol{y}), \boldsymbol{x} - \boldsymbol{y} \rangle$.*

A Bregman distance $D_h$ reduces to the Euclidean distance for $h(\boldsymbol{x}) = \frac{1}{2}||\boldsymbol{x}||^2$. $D_h$ is not symmetric in general, except for the case $h(\boldsymbol{x}) = \frac{1}{2}||\boldsymbol{x}||^2$. In addition, $D_h(\boldsymbol{x}, \boldsymbol{y}) \geq 0$ for all $(\boldsymbol{x}, \boldsymbol{y}) \in \operatorname{dom} h \times \operatorname{int} \operatorname{dom} h$, and $D_h(\boldsymbol{x}, \boldsymbol{y}) = 0$ if and only if $\boldsymbol{x} = \boldsymbol{y}$ due to the strict convexity of $h$. Popular examples of $h$ are $h(\boldsymbol{x}) = \sum_{i=1}^{d} |x_i|^p / p$ for $p \geq 2$, a Shannon entropy $h(\boldsymbol{x}) = \sum_{i=1}^{d} x_i \log x_i$, $\operatorname{dom} h = [0, \infty)^d$, and a Burg entropy $h(\boldsymbol{x}) = -\sum_{i=1}^{d} \log x_i$, $\operatorname{dom} h = (0, \infty)^d$. An appropriate choice of $h$ from this (partial) list, especially that captures the geometry of the constraint sets of the problem, has been found useful in many applications (see, *e.g.*, (Bauschke et al., 2016; Beck & Teboulle, 2003; Bolte et al., 2018; Lu et al., 2018)). This paper, however, chooses $h$ not from the

above standard list. Our choice of $h$ in Section 5 is inspired by that of the PDHG (Chambolle & Pock, 2011; Esser et al., 2010; He & Yuan, 2012).

## 2.2 COMPOSITE MINIMAX PROBLEM AND WEAKLY MONOTONE OPERATOR

We are interested in the minimax problem in a form:

$$\min_{\boldsymbol{u} \in \mathbb{R}^{d_u}} \max_{\boldsymbol{v} \in \mathbb{R}^{d_v}} \left\{ \Phi(\boldsymbol{u}, \boldsymbol{v}) := f(\boldsymbol{u}) + \phi(\boldsymbol{u}, \boldsymbol{v}) - g(\boldsymbol{v}) \right\}, \tag{1}$$

which satisfies the following assumption. Let $d := d_u + d_v$.

**Assumption A1.** $f : \mathbb{R}^{d_u} \to (-\infty, \infty]$ *and* $g : \mathbb{R}^{d_v} \to (-\infty, \infty]$ *are closed, proper and convex functions. A function* $\phi : \mathbb{R}^d \to \mathbb{R}$ *is continuously differentiable and has a Lipschitz continuous gradient,* i.e., *there exists* $L_{\boldsymbol{uu}}, L_{\boldsymbol{uv}}, L_{\boldsymbol{vu}}, L_{\boldsymbol{vv}} > 0$ *such that, for all* $\boldsymbol{u}, \bar{\boldsymbol{u}} \in \mathbb{R}^{d_u}$ *and* $\boldsymbol{v}, \bar{\boldsymbol{v}} \in \mathbb{R}^{d_v}$,

$$||\nabla_{\boldsymbol{u}}\phi(\boldsymbol{u}, \boldsymbol{v}) - \nabla_{\boldsymbol{u}}\phi(\bar{\boldsymbol{u}}, \bar{\boldsymbol{v}})|| \leq L_{\boldsymbol{uu}}||\boldsymbol{u} - \bar{\boldsymbol{u}}|| + L_{\boldsymbol{uv}}||\boldsymbol{v} - \bar{\boldsymbol{v}}||,$$
$$||\nabla_{\boldsymbol{v}}\phi(\boldsymbol{u}, \boldsymbol{v}) - \nabla_{\boldsymbol{v}}\phi(\bar{\boldsymbol{u}}, \bar{\boldsymbol{v}})|| \leq L_{\boldsymbol{vu}}||\boldsymbol{u} - \bar{\boldsymbol{u}}|| + L_{\boldsymbol{vv}}||\boldsymbol{v} - \bar{\boldsymbol{v}}||.$$

Both the gradient $\nabla\phi$ and the saddle-gradient of $\phi$, denoted by $\boldsymbol{M}_\phi := (\nabla_{\boldsymbol{u}}\phi, -\nabla_{\boldsymbol{v}}\phi)$, are $L$-Lipschitz continuous (see Appendix A), *i.e.*, there exists $L > 0$ such that $||\boldsymbol{M}_\phi\boldsymbol{x} - \boldsymbol{M}_\phi\boldsymbol{y}|| \leq L||\boldsymbol{x} - \boldsymbol{y}||$ for all $\boldsymbol{x}, \boldsymbol{y} \in \mathbb{R}^d$.

Finding a first-order stationary point $\boldsymbol{x}_* := (\boldsymbol{u}_*, \boldsymbol{v}_*) \in \mathbb{R}^d$ of (1), is equivalent to finding a zero of the following set-valued saddle-subdifferential operator of $\Phi$:

$$\boldsymbol{M} := \boldsymbol{M}_\phi + (\partial f, \partial g) : \mathbb{R}^d \rightrightarrows (-\infty, \infty]^d. \tag{2}$$

Let $X_*(\boldsymbol{M}) := \{\boldsymbol{x}_* : \boldsymbol{0} \in \boldsymbol{M}\boldsymbol{x}_*\}$ be a nonempty solution set. We consider the squared subgradient norm at $\boldsymbol{x}$, $\min_{\boldsymbol{s} \in \boldsymbol{M}\boldsymbol{x}} ||\boldsymbol{s}||^2$, as an optimality criteria, which is standard in nonconvex-nonconcave minimax problems (Diakonikolas et al., 2021; Lee & Kim, 2021; Pethick et al., 2022; 2023). Under Assumption A1, the operator $\boldsymbol{M}$ (2) satisfies the following weakly monotone condition with $\gamma = \hat{L} := \max\{L_{\boldsymbol{uu}}, L_{\boldsymbol{vv}}\}$ (see Appendix B) and is maximal.

**Assumption A2** (Weak monotonicity). *For some* $\gamma \geq 0$, *an operator* $\boldsymbol{M}$ *is* $\gamma$-*weakly monotone,* i.e., $\langle \boldsymbol{x} - \boldsymbol{y}, \boldsymbol{w} - \boldsymbol{z} \rangle \geq -\gamma||\boldsymbol{x} - \boldsymbol{y}||^2$ *for all* $(\boldsymbol{x}, \boldsymbol{w}), (\boldsymbol{y}, \boldsymbol{z}) \in \operatorname{gra} \boldsymbol{M}$, *where* $\operatorname{gra} \boldsymbol{M} := \{(\boldsymbol{x}, \boldsymbol{w}) \in \mathbb{R}^d \times \mathbb{R}^d : \boldsymbol{w} \in \boldsymbol{M}\boldsymbol{x}\}$ *denotes the graph of* $\boldsymbol{M}$. *Also, it is maximal,* i.e., *there exists no* $\gamma$-*weakly monotone operator that its graph properly contains* $\operatorname{gra} \boldsymbol{M}$.

## 2.3 STRUCTURED NONCONVEX-NONCONCAVE PROBLEM

We consider the nonconvex-nonconcave condition in (Diakonikolas et al., 2021), named the weak Minty variational inequality (MVI). (See Appendix E for the strong MVI condition in (Song et al., 2020; Zhou et al., 2017) and its related analysis.) The MVI problem is to find $\boldsymbol{x}_*$ such that $\langle \boldsymbol{x} - \boldsymbol{x}_*, \boldsymbol{w} \rangle \geq 0$ for all $(\boldsymbol{x}, \boldsymbol{w}) \in \operatorname{gra} \boldsymbol{M}$. For a continuous $\boldsymbol{M}$, a solution set of the MVI problem is a subset of $X_*(\boldsymbol{M})$, and if $\boldsymbol{M}$ is monotone, they are equivalent. The MVI condition, assuming that a solution of the MVI problem exists, is studied in (Dang & Lan, 2015; Malitsky, 2020), which is also studied under the name, the coherence, in (Mertikopoulos et al., 2019; Song et al., 2020; Zhou et al., 2017). Recently, Diakonikolas et al. (2021) introduced the following weaker condition, named weak MVI condition.

**Assumption A3** (Weak MVI). *For some* $\rho \geq 0$, *there exists a solution* $\boldsymbol{x}_* \in X_*(\boldsymbol{M})$ *such that*

$$\langle \boldsymbol{x} - \boldsymbol{x}_*, \boldsymbol{w} \rangle \geq -\frac{\rho}{2}||\boldsymbol{w}||^2, \quad \forall (\boldsymbol{x}, \boldsymbol{w}) \in \operatorname{gra} \boldsymbol{M}.$$

Let $X_*^\rho(\boldsymbol{M})$ be the associated solution set. Assumption A3 is implied by the $-\frac{\rho}{2}$-comonotononicity (Bauschke et al., 2021), or equivalently the $\frac{\rho}{2}$-cohypomonotonicity (Combettes & Pennanen, 2004), *i.e.*, $\langle \boldsymbol{x} - \boldsymbol{y}, \boldsymbol{w} - \boldsymbol{z} \rangle \geq -\frac{\rho}{2}||\boldsymbol{w} - \boldsymbol{z}||^2$ for all $(\boldsymbol{x}, \boldsymbol{w}), (\boldsymbol{y}, \boldsymbol{z}) \in \operatorname{gra} \boldsymbol{M}$. The comonotonicity is also implied by the $\alpha \geq 0$-interaction dominant condition in (Grimmer et al., 2022) (see (Lee & Kim, 2021, Example 1)). Daskalakis et al. (2020, Proposition 2) and Diakonikolas et al. (2021) consider a (constrained) two-agent zero-sum reinforcement learning problem, called the von Neumann ratio game, that satisfies the weak MVI condition, but neither the MVI condition nor the comonotonicity condition. Pethick et al. (2022) also provides examples satisfying the weak MVI.

## 3 EXISTING GRADIENT-BASED METHODS

**Extragradient methods**    Unlike gradient descent in minimization, gradient descent ascent method diverges even for a simple bilinear minimax problem (Mescheder et al., 2018; Zhang & Yu, 2020). Therefore, extragradient-type methods (Hsieh et al., 2019; Korpelevich, 1976; Malitsky, 2020; Nesterov, 2007; Popov, 1980) with a better convergence behavior have gained interest. Especially, the extragradient method works under the MVI condition (Dang & Lan, 2015; Mertikopoulos et al., 2019), and recently, its variants, named EG+ (Diakonikolas et al., 2021) and CEG+ (Pethick et al., 2022), have been found to work under the weak MVI condition.

**Primal-dual hybrid gradient method**    For a bilinearly-coupled convex-concave problem $\min_{\boldsymbol{u}} \max_{\boldsymbol{v}} \{f(\boldsymbol{u}) + \langle \boldsymbol{u},\, \boldsymbol{Bv} \rangle - g(\boldsymbol{v})\}$, which is an instance of (1), the primal-dual hybrid gradient (PDHG) (Chambolle & Pock, 2011; Esser et al., 2010), for $k = 0, 1, \ldots,$

$$\boldsymbol{u}_{k+1} = \text{prox}_{\tau f}[\boldsymbol{u}_k - \tau \boldsymbol{Bv}_k],$$
$$\boldsymbol{v}_{k+1} = \text{prox}_{\tau g}[\boldsymbol{v}_k + \tau \boldsymbol{B}^\top (2\boldsymbol{u}_{k+1} - \boldsymbol{u}_k)],$$

is most widely used, where the proximal operator is defined as $\text{prox}_\psi := (\boldsymbol{I} + \partial \psi)^{-1}$. The PDHG is similar to the alternating (proximal) gradient descent ascent method, except that the PDHG uses the term $\boldsymbol{B}^\top (2\boldsymbol{u}_{k+1} - \boldsymbol{u}_k)$ in the $\boldsymbol{v}_{k+1}$ update, instead of $\boldsymbol{B}^\top \boldsymbol{u}_{k+1}$. This simple modification improves convergence, which is a one-sided variant of the extragradient-type methods, while being practically superior to the plain extragradient in a bilinear problem (Chambolle & Pock, 2011).

In (He & Yuan, 2012), the PDHG is shown to be equivalent to the *preconditioned* proximal point method with a (linear) preconditioner $\boldsymbol{P}(\boldsymbol{u}, \boldsymbol{v}) := (\frac{1}{\tau}\boldsymbol{u} - \boldsymbol{Bv}, -\boldsymbol{B}^\top \boldsymbol{u} + \frac{1}{\tau}\boldsymbol{v})$. In Section 5, we extend this understanding to a nonlinear problem. Note that the PDHG has been extended to tackle convex-concave problems (1) in (Hamedani & Aybat, 2021; Yadav et al., 2018; Zhao, 2019) by directly generalizing the term $\boldsymbol{B}^\top (2\boldsymbol{u}_{k+1} - \boldsymbol{u}_k)$. In particular, the prediction method in (Yadav et al., 2018) uses the term $\nabla_{\boldsymbol{v}}\phi(2\boldsymbol{u}_{k+1} - \boldsymbol{u}_k, \boldsymbol{v}_k)$, and Zhao (2019) and Hamedani & Aybat (2021) consider $2\nabla_{\boldsymbol{v}}\phi(\boldsymbol{u}_k, \boldsymbol{v}_k) - \nabla_{\boldsymbol{v}}\phi(\boldsymbol{u}_{k-1}, \boldsymbol{v}_{k-1})$ with different update ordering. Section 5 proposes a different extension of the PDHG method, by nonlinearizing the preconditioner $\boldsymbol{P}$, leading to the *semi*-implicit gradient term $2\nabla_{\boldsymbol{v}}\phi(\boldsymbol{u}_{k+1}, \boldsymbol{v}_{k+1}) - \nabla_{\boldsymbol{v}}\phi(\boldsymbol{u}_k, \boldsymbol{v}_k)$.

**Gradient descent with max-oracle**    The proposed SA gradient method resembles the gradient descent with max-oracle (GDmax), so we discuss it here. Consider a constrained minimax problem $\min_{\boldsymbol{u} \in \mathcal{U}} \max_{\boldsymbol{v} \in \mathcal{V}} \phi(\boldsymbol{u}, \boldsymbol{v})$, an instance of (1). Then, the GDmax solves the equivalent minimization problem $\min_{\boldsymbol{u} \in \mathcal{U}} \{\Psi(\boldsymbol{u}) := \max_{\boldsymbol{v} \in \mathcal{V}} \phi(\boldsymbol{u}, \boldsymbol{v})\}$ by a gradient descent method on $\boldsymbol{u}$ (Barazandeh & Razaviyayn, 2020; Jin et al., 2020; Nouiehed et al., 2019). This requires a gradient computation $\nabla\Psi(\boldsymbol{u}) = \nabla_{\boldsymbol{u}}\phi(\boldsymbol{u}, \boldsymbol{v}_*(\boldsymbol{u}))$, for $\boldsymbol{v}_*(\boldsymbol{u}) := \arg\max_{\boldsymbol{v} \in \mathcal{V}} \phi(\boldsymbol{u}, \boldsymbol{v})$, based on Danskin's theorem (Danskin, 1967). This involves a maximization with respect to $\boldsymbol{v}$. Unfortunately, even for a smooth $\phi$, the function $\Psi$ is not differentiable in general. If a smooth function $\phi$ is further assumed to be strongly concave on $\boldsymbol{v}$ with a convex compact set $\mathcal{V}$, the function $\Psi$ is differentiable and one can apply a gradient descent method on $\boldsymbol{u}$ (Barazandeh & Razaviyayn, 2020; Nouiehed et al., 2019). In many practical cases, the strong concavity is not given, so regularization can be useful for the GDmax, but this requires tuning parameters and makes it unable to find an exact solution of the original problem.

Table 1 compares the convergence guarantees of the extragradient-type, the PDHG-type, the GDmax-type and the proposed SA gradient methods. In particular, the SA gradient converges under settings that the extragradient-type methods work, unlike the other two types, which is our major contribution.

## 4 BREGMAN PROXIMAL POINT (BPP) METHOD

### 4.1 THE $h$-RESOLVENT

The $h$-resolvent of a monotone operator $\boldsymbol{M}$ with respect to a Legendre function $h$ is defined as $\boldsymbol{R}_{\boldsymbol{M}}^h := (\nabla h + \boldsymbol{M})^{-1} \nabla h$ (Eckstein, 1993), where we omit $\boldsymbol{M}$ and $h$ in $\boldsymbol{R}_{\boldsymbol{M}}^h$ for simplicity hereafter, unless necessary. This reduces to the standard resolvent operator $(\boldsymbol{I} + \boldsymbol{M})^{-1}$ for $h = \frac{1}{2}||\cdot||^2$, where $\boldsymbol{I}$ is an identity operator. The $h$-resolvent $\boldsymbol{R}$ is single-valued on its domain for a monotone operator $\boldsymbol{M}$ (Bauschke et al., 2003, Proposition 3.8), and we extend this for a weakly monotone operator.

| Method | | Monotone | MVI | Weak MVI | Optimality |
|---|---|---|---|---|---|
| Type | Name | | ($\rho = 0$) | ($\rho > 0$) | Measure |
| Extragradient | EG, DE[†] | ✓ | | | |
| | EG, OptDE[‡] | ✓ | ✓ | | |
| | EG+,CEG+[§] | ✓ | ✓ | ✓ | Squared gradient norm |
| PDHG | PDHG[⋆] Prediction[∘] | ✓ | | | |
| GDmax | GDmax[*] | | | | |
| Semi-anchoring | SA-GDmax | ✓ | ✓ | ✓ | Bregman distance (6) |

Table 1: Comparison of the problem settings for the extragradient-type, PDHG-type, GDmax-type and the proposed methods; The SA-GDmax generalizes both the PDHG and GDmax, and converges for the settings that the EG+/CEG+ work. In addition, EG+/CEG+ and SA-GDmax have the same $O(1/k)$ worst-case rate, while the latter is in terms of a specific Bregman distance (6) that upper bounds the squared gradient norm used for the rate of the former. [†](Korpelevich, 1976; Nemirovski, 2004; Nesterov, 2007), [‡](Dang & Lan, 2015; Mertikopoulos et al., 2019; Song et al., 2020), [§](Diakonikolas et al., 2021; Pethick et al., 2022) [⋆](Chambolle & Pock, 2011; Esser et al., 2010), [∘](Hamedani & Aybat, 2021; Yadav et al., 2018; Zhao, 2019), [*](Nouiehed et al., 2019; Jin et al., 2020)

**Lemma 1.** *Let $M$ satisfy Assumption A2 for some $\gamma \geq 0$, and $h$ be a $\mu_h$-strongly convex Legendre function. Then, if $\mu_h > \gamma$, the $h$-resolvent $R$ is single-valued on* int dom $h$.

### 4.2 BPP UNDER WEAK MVI CONDITION

The BPP method (Eckstein, 1993) iteratively applies the $h$-resolvent as, for $k = 0, 1, \ldots$,

$$\boldsymbol{x}_{k+1} = \boldsymbol{R}(\boldsymbol{x}_k), \tag{3}$$

which converges to a zero of a monotone operator $\boldsymbol{M}$. We analyze the worst-case convergence behavior of the BPP under Assumptions A2 and A3. We first state the Bregman nonexpansivity of $\boldsymbol{R}$ below. For $\rho = 0$ and for any Legendre $h$, this reduces to the quasi-Bregman firmly nonexpansive property $D_h(\boldsymbol{x}_*, \boldsymbol{R}\boldsymbol{x}) \leq D_h(\boldsymbol{x}_*, \boldsymbol{x}) - D_h(\boldsymbol{R}\boldsymbol{x}, \boldsymbol{x})$ (Borwein et al., 2011; Eckstein, 1993).

**Lemma 2.** *Let $M$ satisfy Assumption A3 for some $\rho \geq 0$, and $h$ be an $L_h$-smooth Legendre function. Then, if $\boldsymbol{R}\boldsymbol{x}$ exists, for any $\boldsymbol{x}_* \in X_*^\rho(\boldsymbol{M})$, $D_h(\boldsymbol{x}_*, \boldsymbol{R}\boldsymbol{x}) \leq D_h(\boldsymbol{x}_*, \boldsymbol{x}) - (1 - \rho L_h)D_h(\boldsymbol{R}\boldsymbol{x}, \boldsymbol{x})$.*

This lemma presents that the condition $\rho L_h \leq 1$ guarantees the quasi-Bregman nonexpansiveness $D_h(\boldsymbol{x}_*, \boldsymbol{R}\boldsymbol{x}) \leq D_h(\boldsymbol{x}_*, \boldsymbol{x})$. We then have the following worst-case rate in terms of the best Bregman distance between two successive iterates (consequently the best squared subgradient norm) under the weak MVI condition, and the convergence property of the iterate sequences. These built upon (Eckstein, 1993, Theorem 1) of the BPP for a monotone operator.

**Theorem 1.** *Let $M$ satisfy Assumptions A2 and A3 for some $\gamma, \rho \geq 0$, and $h$ be a $\mu_h$-strongly convex and $L_h$-smooth Legendre function with $\mu_h > \gamma$ and $\rho L_h < 1$, respectively. Then, the sequence $\{\boldsymbol{x}_k\}$ of the BPP method (3) satisfies, for $k \geq 1$ and for any $\boldsymbol{x}_* \in X_*^\rho(\boldsymbol{M})$,*

$$\min_{i=1,\ldots,k} \min_{\boldsymbol{s}_i \in \boldsymbol{M}\boldsymbol{x}_i} \frac{||\boldsymbol{s}_i||^2}{2L_h} \leq \min_{i=1,\ldots,k} D_h(\boldsymbol{x}_i, \boldsymbol{x}_{i-1}) \leq \frac{D_h(\boldsymbol{x}_*, \boldsymbol{x}_0)}{(1 - \rho L_h)\,k}.$$

*Moreover, all limit points of the sequence $\{\boldsymbol{x}_k\}$ are in $X_*(\boldsymbol{M})$, and if we further assume that $X_*^\rho(\boldsymbol{M}) = X_*(\boldsymbol{M})$, the sequence $\{\boldsymbol{x}_k\}$ converges to a solution $\boldsymbol{x}_* \in X_*^\rho(\boldsymbol{M})$.*

### 4.3 BPP WITH PROJECTION ONTO A SEPARATING HYPERPLANE

We further develop a variant of the BPP that iteratively projects a point onto the following hyperplane $H(\boldsymbol{x}) := \left\{ \bar{\boldsymbol{x}} \in \mathbb{R}^d \ : \ \langle \nabla h(\boldsymbol{x}) - \nabla h(\boldsymbol{R}\boldsymbol{x}), \boldsymbol{x} - \bar{\boldsymbol{x}} \rangle = \left( \frac{1}{L_h} - \frac{\rho}{2} \right) ||\nabla h(\boldsymbol{x}) - \nabla h(\boldsymbol{R}\boldsymbol{x})||^2 \right\}$ that separates $\boldsymbol{x}$ and $X_*^\rho(\boldsymbol{M})$ under the condition $\rho L_h < 2$ (unless $\nabla h(\boldsymbol{x}) = \nabla h(\boldsymbol{R}\boldsymbol{x})$), for any $\boldsymbol{x} \in \mathbb{R}^d$ (see Appendix C.4). We leave investigating other choices of separating hyperplane as future work.

The corresponding BPP with projection updates as

$$\boldsymbol{x}_{k+1} = P_{H(\boldsymbol{x}_k)}(\boldsymbol{x}_k) = \boldsymbol{x}_k - \left(\frac{1}{L_h} - \frac{\rho}{2}\right)(\nabla h(\boldsymbol{x}_k) - \nabla h(\boldsymbol{R}\boldsymbol{x}_k)). \tag{4}$$

This requires additional computation of $\nabla h$, compared to the standard BPP, while generating a quasi-firmly nonexpansive sequence, as shown in Lemma 3. Note that such projection technique, originally appeared in (Solodov & Svaiter, 1999), has been also used for the extragradient method in (Pethick et al., 2022), yielding a convergence guarantee for a larger range of $\rho$.

**Lemma 3.** *Let $\boldsymbol{M}$ satisfy Assumption A3 for some $\rho \geq 0$, and $h$ be an $L_h$-smooth Legendre function with $\rho L_h \leq 2$. Then, if $\boldsymbol{R}\boldsymbol{x}$ exists, for any $\boldsymbol{x}_* \in X_*^\rho(\boldsymbol{M})$, $||\tilde{\boldsymbol{R}}\boldsymbol{x} - \boldsymbol{x}_*||^2 \leq ||\boldsymbol{x} - \boldsymbol{x}_*||^2 - ||\tilde{\boldsymbol{R}}\boldsymbol{x} - \boldsymbol{x}||^2$, where $\tilde{\boldsymbol{R}} := \boldsymbol{I} - \left(\frac{1}{L_h} - \frac{\rho}{2}\right)(\nabla h - \nabla h \boldsymbol{R})$.*

We then have the following worst-case rate in terms of the squared subgradient norm, and the convergence property of the iterate sequences, for a larger range of $\rho$, compared to the standard BPP.

**Theorem 2.** *Let $\boldsymbol{M}$ satisfy Assumptions A2 and A3 for some $\gamma, \rho \geq 0$, and $h$ be a $\mu_h$-strongly convex and $L_h$-smooth Legendre function with $\mu_h > \gamma$ and $\rho L_h < 2$, respectively. Then, the sequence $\{\boldsymbol{x}_k\}$ of the BPP method with projection (4) satisfies, for $k \geq 1$ and for any $\boldsymbol{x}_* \in X_*^\rho(\boldsymbol{M})$,*

$$\min_{i=1,\ldots,k} \min_{\boldsymbol{s}_i \in \boldsymbol{M}\boldsymbol{R}\boldsymbol{x}_{i-1}} \frac{||\boldsymbol{s}_i||^2}{2L_h} \leq \frac{2L_h||\boldsymbol{x}_0 - \boldsymbol{x}_*||^2}{(2 - \rho L_h)^2 k}.$$

*Moreover, all limit points of the sequence $\{\boldsymbol{x}_k\}$ are in $X_*(\boldsymbol{M})$, and if we further assume that $X_*^\rho(\boldsymbol{M}) = X_*(\boldsymbol{M})$, the sequence $\{\boldsymbol{x}_k\}$ converges to a solution $\boldsymbol{x}_* \in X_*^\rho(\boldsymbol{M})$.*

## 5 SEMI-ANCHORED GRADIENT METHODS

### 5.1 CONSTRUCTING SEMI-ANCHORED GRADIENT METHOD FROM PDHG VIA BPP

Inspired by the linear preconditioner $\boldsymbol{P}(\boldsymbol{u}, \boldsymbol{v})$ of the PDHG, which is equivalent to choosing $h(\boldsymbol{u}, \boldsymbol{v}) = \frac{1}{2\tau}\left(||\boldsymbol{u}||^2 + ||\boldsymbol{v}||^2\right) - \langle \boldsymbol{u}, \boldsymbol{B}\boldsymbol{v}\rangle$ in the BPP, our main contribution of this paper is to consider the following Legendre function:

$$h(\boldsymbol{u}, \boldsymbol{v}) = \frac{1}{2\tau}\left(||\boldsymbol{u}||^2 + ||\boldsymbol{v}||^2\right) - \phi(\boldsymbol{u}, \boldsymbol{v}). \tag{5}$$

Under Assumption A1, this $h$ is $\left(\frac{1}{\tau} - L\right)$-strongly convex[1] when $\frac{1}{\tau} > L$. This yields the Bregman distance, which is a difference between the function $\phi$ and its quadratic upper bound at $\boldsymbol{y}$:

$$D_h(\boldsymbol{x}, \boldsymbol{y}) = \phi(\boldsymbol{y}) + \langle \nabla \phi(\boldsymbol{y}), \boldsymbol{x} - \boldsymbol{y}\rangle + \frac{1}{2\tau}||\boldsymbol{x} - \boldsymbol{y}||^2 - \phi(\boldsymbol{x}). \tag{6}$$

This Bregman distance $D_h(\boldsymbol{x}_k, \boldsymbol{x}_{k-1})$ of the successive iterates is used as an optimality measure in our worst-case rate analysis, which has not been observed anywhere in minimax optimization literatures. We believe having such optimality measure, tailored from using the analysis of the Bregman proximal point method and our specific choice of $h$, may be the key of having successful extension of the PDHG to nonconvex-nonconcave problems, which calls for further investigation. We have already shown in Theorem 1 that this upper bounds the squared subgradient norm, a standard optimality measure, so our rate analysis easily translates to standard rate analysis.

We are now ready to state our new method. Since $\boldsymbol{M}$ in (2) is $\hat{L} = \max\{L_{\boldsymbol{u}\boldsymbol{u}}, L_{\boldsymbol{v}\boldsymbol{v}}\}$-weakly monotone, the corresponding BPP update (3) with $\nabla h = \frac{1}{\tau}\boldsymbol{I} - \nabla\phi$ is well-defined for $\frac{1}{\tau} - L > \hat{L}$, by Lemma 1. The BPP update (3) with $h$ in (5) is $(\boldsymbol{u}_{k+1}, \boldsymbol{v}_{k+1}) = \left(\frac{1}{\tau}\boldsymbol{I} - \nabla\phi + \boldsymbol{M}\right)^{-1}\left(\frac{1}{\tau}\boldsymbol{I} - \nabla\phi\right)(\boldsymbol{u}_k, \boldsymbol{v}_k)$. Rewriting this in the minimization and maximization form respectively leads to

$$\boldsymbol{u}_{k+1} = \arg\min_{\boldsymbol{u} \in \mathbb{R}^{d_u}} \left\{\frac{1}{2\tau}||\boldsymbol{u} - (\boldsymbol{u}_k - \tau\nabla_{\boldsymbol{u}}\phi(\boldsymbol{u}_k, \boldsymbol{v}_k))||^2 + f(\boldsymbol{u})\right\} \tag{7}$$

---

[1]We have that $\langle \nabla h(\boldsymbol{x}) - \nabla h(\boldsymbol{y}), \boldsymbol{x} - \boldsymbol{y}\rangle = \frac{1}{\tau}||\boldsymbol{x} - \boldsymbol{y}||^2 - \langle \nabla\phi(\boldsymbol{x}) - \nabla\phi(\boldsymbol{y}), \boldsymbol{x} - \boldsymbol{y}\rangle \geq \left(\frac{1}{\tau} - L\right)||\boldsymbol{x} - \boldsymbol{y}||^2$ for all $\boldsymbol{x}, \boldsymbol{y} \in \mathbb{R}^d$, where the last inequality uses the $L$-Lipschitz continuity of $\nabla\phi$.

$$\boldsymbol{v}_{k+1} = \arg\max_{\boldsymbol{v} \in \mathbb{R}^{d_v}} \left\{ 2\phi(\boldsymbol{u}_{k+1}, \boldsymbol{v}) - \frac{1}{2\tau} ||\boldsymbol{v} - (\boldsymbol{v}_k - \tau \nabla_{\boldsymbol{v}} \phi(\boldsymbol{u}_k, \boldsymbol{v}_k))||^2 - g(\boldsymbol{v}) \right\}.$$

The minimization in $\boldsymbol{u}$ can be solved by one proximal gradient update, while the maximization in $\boldsymbol{v}$ is equivalent to an implicit update $\boldsymbol{v}_{k+1} = \text{prox}_{\tau g}[\boldsymbol{v}_k + \tau(2\nabla_{\boldsymbol{v}}\phi(\boldsymbol{u}_{k+1}, \boldsymbol{v}_{k+1}) - \nabla_{\boldsymbol{v}}\phi(\boldsymbol{u}_k, \boldsymbol{v}_k))]$. Since the proximal point, *i.e.,* anchoring, only happens in $\boldsymbol{v}$, we name this technique to be semi-anchoring (SA). In particular, when equipped with the exact maximization oracle, we call this method, SA-GDmax, as it resembles GDmax. Similarly, built upon the BPP with projection, we obtain SA-GDmax with projection.

Lemma 1 and Theorems 1 and 2 of the BPP method (with projection), that will be used for the rate analysis of SA-GDmax, also apply to the standard proximal point method (with $h(\boldsymbol{x}) = \frac{1}{2\tau}||\boldsymbol{x}||^2$), *i.e.,* $\boldsymbol{x}_{k+1} = (I + \tau \boldsymbol{M})^{-1}\boldsymbol{x}_k$. Such, however, requires solving a regularized *minimax* $(\boldsymbol{u}_{k+1}, \boldsymbol{v}_{k+1}) = \arg\min_{\boldsymbol{u}} \max_{\boldsymbol{v}} \left\{ f(\boldsymbol{u}) + \phi(\boldsymbol{u}, \boldsymbol{v}) - g(\boldsymbol{v}) + \frac{1}{2\tau}||\boldsymbol{u} - \boldsymbol{u}_k||^2 - \frac{1}{2\tau}||\boldsymbol{v} - \boldsymbol{v}_k||^2 \right\}$, at each iteration, while the SA-GDmax needs one gradient descent step and a *maximization* at each iteration. Both methods intrinsically have an implicit regularization (smoothing), and thus have a good convergence guarantee, while the latter is preferred in terms of the computational complexity.

## 5.2 DEVELOPING A PRACTICAL VARIANT OF SEMI-ANCHORED GRADIENT METHOD

In many practical cases, exact maximization oracle is not available. So as a first attempt to making the SA gradient method more practical, we consider applying an iterative method for the maximization in $\boldsymbol{v}$. The maximization problem in $\boldsymbol{v}$ consists of a $\left(\frac{1}{\tau} + 2L_{\boldsymbol{vv}}\right)$-smooth and $\left(\frac{1}{\tau} - 2L_{\boldsymbol{vv}}\right)$-strongly concave function, and a concave but possibly nonsmooth function $-g$. We can thus use total $J$ number of (inner) proximal gradient steps (using the Lipschitz continuity of $\nabla_{\boldsymbol{vv}}\phi(\boldsymbol{u}_{k+1}, \cdot)$) for the (inexact) maximization in $\boldsymbol{v}$; fast proximal gradient methods (Beck & Teboulle, 2009; Chambolle & Pock, 2016) can be used for acceleration. Since this involves multiple gradient steps, we name this semi-anchored multi-step gradient descent ascent (SA-MGDA). The resulting SA-MGDA method (with projection) is illustrated in Algorithm 1 (for $L_{\boldsymbol{vv}} > 0$). The $\boldsymbol{v}_{k,j+1}$ update involves a convex combination of an *anchor* point, $\boldsymbol{v}_k - \tau\nabla_{\boldsymbol{v}}\phi(\boldsymbol{u}_k, \boldsymbol{v}_k)$, that only depends on the previous point $(\boldsymbol{u}_k, \boldsymbol{v}_k)$ and a recent point, $\boldsymbol{v}_{k,j} + \frac{1}{L_{\boldsymbol{vv}}}\nabla_{\boldsymbol{v}}\phi(\boldsymbol{u}_{k+1}, \boldsymbol{v}_{k,j})$, that depends on $(\boldsymbol{u}_{k+1}, \boldsymbol{v}_{k,j})$. So the name anchoring is more apparent here.

---

**Algorithm 1** SA-MGDA (with projection) for (1) with $L_{\boldsymbol{vv}} > 0$

---

**Input:** $\boldsymbol{x}_0 = (\boldsymbol{u}_0, \boldsymbol{v}_0), \tau, \eta = \frac{\tau}{1+2L_{\boldsymbol{vv}}\tau}$
**for** $k = 0, 1, \ldots$ **do**
  $\hat{\boldsymbol{u}}_{k+1} = \text{prox}_{\tau f}[\boldsymbol{u}_k - \tau\nabla_{\boldsymbol{u}}\phi(\boldsymbol{u}_k, \boldsymbol{v}_k)], \quad \boldsymbol{v}_{k,0} = \boldsymbol{v}_k$
  **for** $j = 0, \ldots, J - 1$ **do**
    $\boldsymbol{v}_{k,j+1} = \text{prox}_{\eta g} \left[ \frac{\eta}{\tau}(\boldsymbol{v}_k - \tau\nabla_{\boldsymbol{v}}\phi(\boldsymbol{u}_k, \boldsymbol{v}_k)) + 2\eta L_{\boldsymbol{vv}}(\boldsymbol{v}_{k,j} + \frac{1}{L_{\boldsymbol{vv}}}\nabla_{\boldsymbol{v}}\phi(\hat{\boldsymbol{u}}_{k+1}, \boldsymbol{v}_{k,j})) \right]$
  $\hat{\boldsymbol{v}}_{k+1} = \boldsymbol{v}_{k,J}, \quad \hat{\boldsymbol{x}}_{k+1} = (\hat{\boldsymbol{u}}_{k+1}, \hat{\boldsymbol{v}}_{k+1})$
  **if** SA-MGDA **then** $\boldsymbol{x}_{k+1} = \hat{\boldsymbol{x}}_{k+1}$
  **else if** SA-MGDA with projection **then** $\boldsymbol{x}_{k+1} = \boldsymbol{x}_k - \left(\frac{1}{\frac{1}{\tau}+L} - \frac{\rho}{2}\right)(\boldsymbol{M}_\phi\hat{\boldsymbol{x}}_{k+1} - \boldsymbol{M}_\phi\boldsymbol{x}_k)$

---

We leave making a more practical version of the SA gradient method, without both exact maximization oracle and inner iterations, as future work. Note that the extragradient method is such a practical variant of the proximal point method. We believe that our development of the SA-GDmax and the SA-MGDA can be a foundation for further development of a practical PDHG-like method that is comparable or outperforming the extragradient method in nonconvex-nonconcave minimax problems. We next provide worst-case rates of the SA-GDmax and SA-MGDA that are comparable to those of the extragradients.

## 6 ANALYZING SEMI-ANCHORED GRADIENT METHODS

### 6.1 ANALYZING THE EXACT SA GRADIENT METHOD: SA-GDMAX

The worst-case rate of the SA-GDmax directly follows from Lemma 1 and Theorem 1 of the BPP method, for a specific $h$ in (5) that is $\mu_h$-strongly convex and $L_h$-smooth with $\mu_h = \frac{1}{\tau} - L$ and

$L_h = \frac{1}{\tau} + L$. In specific, the proof follows from Theorem 1 with constraints $\mu_h > \hat{L}$ and $\rho L_h < 1$, yielding $\tau < 1/(L+\hat{L})$ and $\tau > \rho/(1-\rho L)$. We also need $\rho < 1/(2L+\hat{L})$, so that $\tau$ exists.

**Theorem 3.** *Let $M$ (2) of the composite problem (1) satisfy Assumption A3 for some $\rho \in \left[0, \frac{1}{2L+\hat{L}}\right)$, and let $f, g$ and $\phi$ satisfy Assumption A1. Then, the sequence $\{x_k\}$ of the SA-GDmax (i.e., SA-MGDA with $J = \infty$) satisfies, for $k \geq 1$, $\tau \in \left(\frac{\rho}{1-\rho L}, \frac{1}{L+\hat{L}}\right)$ and for any $x_* \in X_*^\rho(M)$,*

$$\min_{i=1,\dots,k} \min_{s_i \in Mx_i} \frac{\|s_i\|^2}{2\left(\frac{1}{\tau}+L\right)} \leq \min_{i=1,\dots,k} D_h(x_i, x_{i-1}) \leq \frac{D_h(x_*, x_0)}{\left(1 - \rho\left(\frac{1}{\tau}+L\right)\right)k}.$$

*Moreover, all limit points of the sequence $\{x_k\}$ are in $X_*(M)$, and if we further assume that $X_*^\rho(M) = X_*(M)$, the sequence $\{x_k\}$ converges to a solution $x_* \in X_*^\rho(M)$.*

The EG+ (Diakonikolas et al., 2021) and CEG+ (Pethick et al., 2022) have shown that they have the same $O(1/k)$ rate for the squared gradient norm. We want to note that our result is on the Bregman distance that is an upper bound on the squared gradient norm. So, it is possible that we have a gain in the convergence rate, which we observe in our experiments. Of course, the SA-GDmax is comparable to the extragradient in terms of the computational complexity only if we have a computationally cheap exact maximization oracle.

Similarly, we have the following theorem of the SA-GDmax with projection. With additional computation of $M_\phi \hat{x}_{k+1}$, it works for a larger range of $\rho$ values. The proof follows from Theorem 2 with constraints $\mu_h > \hat{L}$ and $\rho L_h < 2$, yielding $\tau < 1/(L+\hat{L})$ and $\tau > \rho/(2-\rho L)$. We also need $\rho < 2/(2L+\hat{L})$, so that $\tau$ exists. Note that this method becomes more useful for a finite $J$ soon.

**Theorem 4.** *Let $M$ (2) of the composite problem (1) satisfy Assumption A3 for some $\rho \in \left[0, \frac{2}{2L+\hat{L}}\right)$, and let $f, g$ and $\phi$ satisfy Assumption A1. Then, the sequence $\{x_k\}$ of the SA-GDmax with projection satisfies, for $k \geq 1$, $\tau \in \left(\frac{\rho}{2-\rho L}, \frac{1}{L+\hat{L}}\right)$ and for any $x_* \in X_*^\rho(M)$,*

$$\min_{i=1,\dots,k} \min_{s_i \in MRx_{i-1}} \frac{\|s_i\|^2}{2\left(\frac{1}{\tau}+L\right)} \leq \frac{2\left(\frac{1}{\tau}+L\right)\|x_0 - x_*\|^2}{\left(2 - \rho\left(\frac{1}{\tau}+L\right)\right)^2 k}.$$

*Moreover, all limit points of the sequence $\{x_k\}$ are in $X_*(M)$, and if we further assume that $X_*^\rho(M) = X_*(M)$, the sequence $\{x_k\}$ converges to a solution $x_* \in X_*^\rho(M)$.*

## 6.2 ANALYZING THE INEXACT SA GRADIENT METHOD: SA-MGDA

This section studies the convergence behavior of the SA-MGDA (with projection). The SA-MGDA can be viewed as an inexact variant of the BPP method (Eckstein, 1998), which generates a point $x_{k+1}$ different from $Rx_k$ at the $k$th iteration. Therefore, the proof of the following theorem of the SA-MGDA first extends Theorem 1 of the (exact) BPP to its inexact variant in Appendix D.1. Then, we consequently have the following theorem for the SA-MGDA (see Theorem 7 in Appendix D.1 for a more detailed statement). Its gradient computation complexity $O(\epsilon^{-1} \log \epsilon^{-1})$ matches the complexity of the SA-GDmax up to a logarithmic factor. We consider the case where $f$ and $g$ are prox-friendly, so we will neglect the complexity of their proximal operations.

**Theorem 5.** *Let $M$ (2) of the composite problem (1) satisfy Assumption A3 for some $\rho \in \left[0, \frac{1}{2L+\hat{L}}\right)$, and let $f, g$ and $\phi$ satisfy Assumption A1. Then, the SA-MGDA method finds an $\epsilon$-stationary point, i.e., a point $x$ satisfying $\min_{s \in MRx} \frac{\|s\|^2}{2\left(\frac{1}{\tau}+L\right)} \leq D_h(Rx, x) \leq \epsilon$, with $k = O(\epsilon^{-1})$ number of outer iterations and $J = O(\log(\epsilon^{-1}))$ inner iterations, requiring total $O(\epsilon^{-1} \log \epsilon^{-1})$ gradient computations.*

We have exactly the same complexity result for the SA-MGDA with projection, in terms of the squared gradient norm, but for a larger region of $\rho$ values below, illustrating the importance of the projection step. The proof similarly first extends Theorem 2 of the (exact) BPP with projection to its inexact variant in Appendix D.2, and consequently have the following result (see Theorem 8 in Appendix D.2 for a more detailed statement).

**Theorem 6.** *Let $M$ (2) of the composite problem (1) satisfy Assumption A3 for some $\rho \in \left[0, \frac{2}{2L+\hat{L}}\right)$, and let $f, g$ and $\phi$ satisfy Assumption A1. Then, the SA-MGDA with projection finds an $\epsilon$-stationary point, i.e., a point $x$ satisfying $\min_{s \in MRx} \|s\|^2 \leq \epsilon$, with $k = O(\epsilon^{-1})$ number of outer iterations and $J = O(\log(\epsilon^{-1}))$ inner iterations, requiring total $O(\epsilon^{-1} \log \epsilon^{-1})$ gradient computations.*

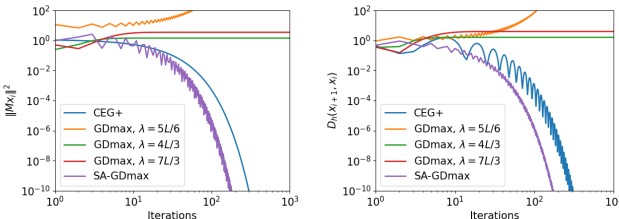

Figure 1: Toy example: (Left) Squared gradient norm, (Right) Our optimality measure (6).

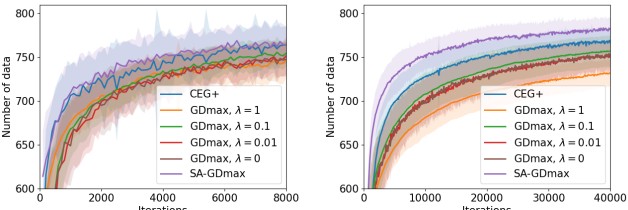

Figure 2: Fair classification: the number of correctly classified test data for the worst category vs. iteration with (Left) $\tau = 0.01$ and (Right) $\tau = 0.001$.

## 7 NUMERICAL RESULTS

We consider two experiments that an efficient max-oracle is available. The first toy example satisfies the weak MVI, while the second one is not known to satisfy it. The latter is provided to see whether the SA works well in practice. For both cases, the SA-GDmax method outperforms the extragradient.

### 7.1 TOY EXAMPLE

We consider a toy example $\phi(u, v) = -\frac{L^2\rho}{4}u^2 + L\sqrt{1 - L^2\rho^2/4}uv + \frac{L^2\rho}{4}v^2$ ($f(u) = g(v) = 0$), which has a saddle operator satisfying $L$-Lipschitz continuity and $\rho$-weak MVI. We compare SA-GDmax with CEG+ and GDmax (to also verify the effectiveness of SA in GDmax). Since GDmax does not converge in our setting, it is applied to a regularized problem $\phi_\lambda(u, v) := \phi(u, v) - \frac{\lambda}{2}\|v - v_0\|^2$ (Nouiehed et al., 2019). Here, we take $\rho = \frac{2}{3L}$, $L = 1$, and $(u_0, v_0) = (1, 1/2)$. Fig. 1 presents that the SA-GDmax outperforms CEG+ with $\bar{\alpha} = \frac{1-L\rho}{2}$, and the GDmax with various choices of $\lambda$.

### 7.2 FAIR CLASSIFICATION

To make the trained model fair to all categories, Mohri et al. (2019) considered a minimax problem that minimizes the maximum loss among the categories. We study such fair classification experiment in Nouiehed et al. (2019) on the Fashion MNIST data set[2] (Xiao et al., 2017). Similar to Nouiehed et al. (2019), we focus on the data labeled as T-shirt/top, Coat, and Shirt. The corresponding minimax problem is $\min_{\boldsymbol{u}} \max_{i=1,2,3} \mathcal{L}_i(\boldsymbol{u})$, where $\boldsymbol{u}$ denotes the parameters of the neural network (see Appendix F for the details), and $\mathcal{L}_1$, $\mathcal{L}_2$, and $\mathcal{L}_3$ denote the cross-entropy losses of the training data in each category, respectively. This is equivalent to $\min_{\boldsymbol{u}} \max_{\boldsymbol{v} \in \mathcal{V}} \sum_{i=1}^3 v_i \mathcal{L}_i(\boldsymbol{u})$, where $\mathcal{V} = \{\boldsymbol{v} \in \mathbb{R}^3_+ : \sum_{i=1}^3 v_i = 1\}$, i.e., $\phi(\boldsymbol{u}, \boldsymbol{v}) = \sum_{i=1}^3 v_i \mathcal{L}_i(\boldsymbol{u})$ with $f = 0$ and $g(\boldsymbol{v}) = \delta_{\mathcal{V}}(\boldsymbol{v})$. Since the problem is not strongly concave in $\boldsymbol{v}$, Nouiehed et al. (2019) applied the GDmax to a regularized problem $\min_{\boldsymbol{u}} \max_{\boldsymbol{v} \in \mathcal{V}} \sum_{i=1}^3 v_i \mathcal{L}_i(\boldsymbol{u}) - \frac{\lambda}{2} \sum_{i=1}^3 v_i^2$. We ran the CEG+, GDmax, and SA-GDmax methods with the same learning rates $\tau = 0.01, 0.001$. For the GDmax, we considered various regularization parameters $\lambda = 0, 0.01, 0.1, 1$. We performed 50 independent simulations for each case, and, in Fig. 2, we report the mean and standard deviation of the number of correctly classified test data (out of 1000) for the worst[3] category, versus iterations. We have gains on SA-GDmax over CEG+ and GDmax, similar to our toy experiment, as expected.

---

[2]This consists of 28×28 grayscale cloth images of ten categories; 60000 data for training and 10000 for test.
[3]The worst category denotes the smallest number of correctly classified test data among the three categories.

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

## CONTENTS

## A   PROOF OF LIPSCHITZ CONTINUITY OF $\nabla\phi$ AND $\mathbf{M}_\phi$

Let $\boldsymbol{x} := (\boldsymbol{u}, \boldsymbol{v})$ and $\boldsymbol{y} := (\bar{\boldsymbol{u}}, \bar{\boldsymbol{v}})$. Since $\|\nabla\phi(\boldsymbol{u}, \boldsymbol{v}) - \nabla\phi(\bar{\boldsymbol{u}}, \bar{\boldsymbol{v}})\| = \|\boldsymbol{M}_\phi \boldsymbol{x} - \boldsymbol{M}_\phi \boldsymbol{y}\|$, it is enough to show that there exists a constant $L > 0$ such that $\|\nabla\phi(\boldsymbol{u}, \boldsymbol{v}) - \nabla\phi(\bar{\boldsymbol{u}}, \bar{\boldsymbol{v}})\| \leq L\|\boldsymbol{x} - \boldsymbol{y}\|$.

By Assumption A1, we have the following bounds

$$\|\nabla\phi(\boldsymbol{u}, \boldsymbol{v}) - \nabla\phi(\bar{\boldsymbol{u}}, \boldsymbol{v})\|^2 = \|\nabla_{\boldsymbol{u}}\phi(\boldsymbol{u}, \boldsymbol{v}) - \nabla_{\boldsymbol{u}}\phi(\bar{\boldsymbol{u}}, \boldsymbol{v})\|^2 + \|\nabla_{\boldsymbol{v}}\phi(\boldsymbol{u}, \boldsymbol{v}) - \nabla_{\boldsymbol{v}}\phi(\bar{\boldsymbol{u}}, \boldsymbol{v})\|^2$$
$$\leq (L_{\boldsymbol{uu}}^2 + L_{\boldsymbol{vu}}^2)\|\boldsymbol{u} - \bar{\boldsymbol{u}}\|^2,$$
$$\|\nabla\phi(\bar{\boldsymbol{u}}, \boldsymbol{v}) - \nabla\phi(\bar{\boldsymbol{u}}, \bar{\boldsymbol{v}})\|^2 = \|\nabla_{\boldsymbol{u}}\phi(\bar{\boldsymbol{u}}, \boldsymbol{v}) - \nabla_{\boldsymbol{u}}\phi(\bar{\boldsymbol{u}}, \bar{\boldsymbol{v}})\|^2 + \|\nabla_{\boldsymbol{v}}\phi(\bar{\boldsymbol{u}}, \boldsymbol{v}) - \nabla_{\boldsymbol{v}}\phi(\bar{\boldsymbol{u}}, \bar{\boldsymbol{v}})\|^2$$
$$\leq (L_{\boldsymbol{uv}}^2 + L_{\boldsymbol{vv}}^2)\|\boldsymbol{v} - \bar{\boldsymbol{v}}\|^2.$$

Then, we can show the Lipschitz continuity as below.

$$\|\nabla\phi(\boldsymbol{u}, \boldsymbol{v}) - \nabla\phi(\bar{\boldsymbol{u}}, \bar{\boldsymbol{v}})\| \leq \|\nabla\phi(\boldsymbol{u}, \boldsymbol{v}) - \nabla\phi(\bar{\boldsymbol{u}}, \boldsymbol{v})\| + \|\nabla\phi(\bar{\boldsymbol{u}}, \boldsymbol{v}) - \nabla\phi(\bar{\boldsymbol{u}}, \bar{\boldsymbol{v}})\|$$
$$\leq \sqrt{L_{\boldsymbol{uu}}^2 + L_{\boldsymbol{vu}}^2}\|\boldsymbol{u} - \bar{\boldsymbol{u}}\| + \sqrt{L_{\boldsymbol{uv}}^2 + L_{\boldsymbol{vv}}^2}\|\boldsymbol{v} - \bar{\boldsymbol{v}}\|$$
$$\leq \sqrt{L_{\boldsymbol{uu}}^2 + L_{\boldsymbol{vu}}^2 + L_{\boldsymbol{uv}}^2 + L_{\boldsymbol{vv}}^2}\sqrt{\|\boldsymbol{u} - \bar{\boldsymbol{u}}\|^2 + \|\boldsymbol{v} - \bar{\boldsymbol{v}}\|^2}$$
$$(\because \text{Cauchy-Schwarz inequality})$$
$$= \sqrt{L_{\boldsymbol{uu}}^2 + L_{\boldsymbol{vu}}^2 + L_{\boldsymbol{uv}}^2 + L_{\boldsymbol{vv}}^2}\|\boldsymbol{x} - \boldsymbol{y}\|.$$

$\square$

## B   PROOF OF WEAK MONOTONICITY OF $\mathbf{M}$

By Assumption A1, $\phi(\cdot, \boldsymbol{v})$ is $L_{\boldsymbol{uu}}$-weakly convex for fixed $\boldsymbol{v}$, and $-\phi(\boldsymbol{u}, \cdot)$ is $L_{\boldsymbol{vv}}$-weakly convex for fixed $\boldsymbol{u}$. Then, using the weak convexity on $\boldsymbol{u}$, we have

$$\phi(\bar{\boldsymbol{u}}, \boldsymbol{v}) \geq \phi(\boldsymbol{u}, \boldsymbol{v}) + \langle\nabla_{\boldsymbol{u}}\phi(\boldsymbol{u}, \boldsymbol{v}), \bar{\boldsymbol{u}} - \boldsymbol{u}\rangle - \frac{L_{\boldsymbol{uu}}}{2}||\bar{\boldsymbol{u}} - \boldsymbol{u}||^2,$$

$$\phi(\boldsymbol{u}, \bar{\boldsymbol{v}}) \geq \phi(\bar{\boldsymbol{u}}, \bar{\boldsymbol{v}}) + \langle\nabla_{\boldsymbol{u}}\phi(\bar{\boldsymbol{u}}, \bar{\boldsymbol{v}}), \boldsymbol{u} - \bar{\boldsymbol{u}}\rangle - \frac{L_{\boldsymbol{uu}}}{2}||\boldsymbol{u} - \bar{\boldsymbol{u}}||^2,$$

for all $\boldsymbol{u}, \bar{\boldsymbol{u}} \in \mathbb{R}^{d_u}$ and $\boldsymbol{v}, \bar{\boldsymbol{v}} \in \mathbb{R}^{d_v}$. Similarly, using the weak convexity on $\boldsymbol{v}$, we have

$$-\phi(\boldsymbol{u}, \bar{\boldsymbol{v}}) \geq -\phi(\boldsymbol{u}, \boldsymbol{v}) - \langle\nabla_{\boldsymbol{v}}\phi(\boldsymbol{u}, \boldsymbol{v}), \bar{\boldsymbol{v}} - \boldsymbol{v}\rangle - \frac{L_{\boldsymbol{vv}}}{2}||\bar{\boldsymbol{v}} - \boldsymbol{v}||^2,$$

$$-\phi(\bar{\boldsymbol{u}}, \boldsymbol{v}) \geq -\phi(\bar{\boldsymbol{u}}, \bar{\boldsymbol{v}}) - \langle\nabla_{\boldsymbol{v}}\phi(\bar{\boldsymbol{u}}, \bar{\boldsymbol{v}}), \boldsymbol{v} - \bar{\boldsymbol{v}}\rangle - \frac{L_{\boldsymbol{vv}}}{2}||\boldsymbol{v} - \bar{\boldsymbol{v}}||^2.$$

Let $\boldsymbol{x} = (\boldsymbol{u}, \boldsymbol{v})$, $\boldsymbol{y} = (\bar{\boldsymbol{u}}, \bar{\boldsymbol{v}})$, $\boldsymbol{w}_\phi = (\nabla_{\boldsymbol{u}}\phi(\boldsymbol{u}, \boldsymbol{v}), -\nabla_{\boldsymbol{v}}\phi(\boldsymbol{u}, \boldsymbol{v}))$, and $\boldsymbol{z}_\phi = (\nabla_{\boldsymbol{u}}\phi(\bar{\boldsymbol{u}}, \bar{\boldsymbol{v}}), -\nabla_{\boldsymbol{v}}\phi(\bar{\boldsymbol{u}}, \bar{\boldsymbol{v}}))$. Then, summing the above four inequalities yields

$$\langle\boldsymbol{x} - \boldsymbol{y}, \boldsymbol{w} - \boldsymbol{z}\rangle \geq \langle\boldsymbol{x} - \boldsymbol{y}, \boldsymbol{w}_\phi - \boldsymbol{z}_\phi\rangle \geq -L_{\boldsymbol{uu}}||\boldsymbol{u} - \bar{\boldsymbol{u}}||^2 - L_{\boldsymbol{vv}}||\boldsymbol{v} - \bar{\boldsymbol{v}}||^2$$
$$\geq -\max\{L_{\boldsymbol{uu}}, L_{\boldsymbol{vv}}\}||\boldsymbol{x} - \boldsymbol{y}||^2,$$

for all $(\boldsymbol{x}, \boldsymbol{w}), (\boldsymbol{y}, \boldsymbol{z}) \in \text{gra}\,\boldsymbol{M}$, where the first inequality uses the convexity of $f$ and $g$.        $\square$

## C   PROOFS FOR SECTION 4

### C.1   PROOF OF LEMMA 1

Note that $\nabla h + \boldsymbol{M} = (\nabla h - \gamma\boldsymbol{I}) + (\boldsymbol{M} + \gamma\boldsymbol{I})$. From the condition that $\mu_h > \gamma$ and $\boldsymbol{M}$ is $\gamma$-weakly monotone, it is straightforward to show that $h - \frac{\gamma}{2}\|\cdot\|^2$ is a Legendre function and $\boldsymbol{M} + \gamma\boldsymbol{I}$ is maximally monotone. Then Bauschke et al. (2010, Corollary 2.3) shows that $\text{ran}(\nabla h + \boldsymbol{M}) = \mathbb{R}^d$. This implies that $\boldsymbol{R}\boldsymbol{x}$ is nonempty for all $\boldsymbol{x} \in \text{int dom}\,h$. Assume that $\boldsymbol{y}, \boldsymbol{z} \in \boldsymbol{R}\boldsymbol{x}$. Since $\nabla h(\boldsymbol{x}) - \nabla h(\boldsymbol{y}) \in \boldsymbol{M}\boldsymbol{y}$ and $\nabla h(\boldsymbol{x}) - \nabla h(\boldsymbol{z}) \in \boldsymbol{M}\boldsymbol{z}$, we have $-\gamma\|\boldsymbol{y} - \boldsymbol{z}\|^2 \leq -\langle\nabla h(\boldsymbol{y}) - \nabla h(\boldsymbol{z}), \boldsymbol{y} - \boldsymbol{z}\rangle \leq -\mu_h\|\boldsymbol{y} - \boldsymbol{z}\|^2$. So if $\mu_h > \gamma$, the inequality implies that $\boldsymbol{y} = \boldsymbol{z}$.

## C.2 PROOF OF LEMMA 2

By the definition of $\boldsymbol{Rx}$, we have $\nabla h(\boldsymbol{x}) - \nabla h(\boldsymbol{Rx}) \in \boldsymbol{MRx}$. Then, Assumption A3 on $\boldsymbol{M}$ implies that

$$
\begin{aligned}
0 &\leq \langle \nabla h(\boldsymbol{x}) - \nabla h(\boldsymbol{Rx}), \ \boldsymbol{Rx} - \boldsymbol{x}_* \rangle + \frac{\rho}{2} \|\nabla h(\boldsymbol{x}) - \nabla h(\boldsymbol{Rx})\|^2 \\
&= \langle \nabla h(\boldsymbol{x}), \ \boldsymbol{Rx} - \boldsymbol{x} \rangle - \langle \nabla h(\boldsymbol{x}), \ \boldsymbol{x}_* - \boldsymbol{x} \rangle + \langle \nabla h(\boldsymbol{Rx}), \ \boldsymbol{x}_* - \boldsymbol{Rx} \rangle + \frac{\rho}{2} \|\nabla h(\boldsymbol{x}) - \nabla h(\boldsymbol{Rx})\|^2 \\
&= -D_h(\boldsymbol{Rx}, \boldsymbol{x}) + D_h(\boldsymbol{x}_*, \boldsymbol{x}) - D_h(\boldsymbol{x}_*, \boldsymbol{Rx}) + \frac{\rho}{2} \|\nabla h(\boldsymbol{x}) - \nabla h(\boldsymbol{Rx})\|^2 \\
&\leq -D_h(\boldsymbol{Rx}, \boldsymbol{x}) + D_h(\boldsymbol{x}_*, \boldsymbol{x}) - D_h(\boldsymbol{x}_*, \boldsymbol{Rx}) + \rho L_h D_h(\boldsymbol{Rx}, \boldsymbol{x}),
\end{aligned}
$$

where the last inequality follows from the convex and $L_h$-smooth properties of $h$, *i.e.*, $\frac{1}{2L_h}\|\nabla h(\boldsymbol{x}) - \nabla h(\boldsymbol{y})\|^2 \leq D_h(\boldsymbol{x}, \boldsymbol{y})$ for all $\boldsymbol{x}, \boldsymbol{y}$ (Nesterov, 2018, Theroem 2.1.5).

## C.3 PROOF OF THEOREM 1

By Lemma 1, the condition $\mu_h > \gamma$ implies that $\boldsymbol{Rx}$ exists for any $\boldsymbol{x}$. By Lemma 2, we get

$$
D_h(\boldsymbol{x}_*, \boldsymbol{x}_i) \leq D_h(\boldsymbol{x}_*, \boldsymbol{x}_{i-1}) - (1 - \rho L_h) D_h(\boldsymbol{x}_i, \boldsymbol{x}_{i-1}) \tag{8}
$$

for all $i \geq 1$. By summing over the above inequality, we get

$$
\begin{aligned}
\sum_{i=1}^{k} (1 - \rho L_h) D_h(\boldsymbol{x}_i, \boldsymbol{x}_{i-1}) &\leq \sum_{i=1}^{k} \left( D_h(\boldsymbol{x}_*, \boldsymbol{x}_{i-1}) - D_h(\boldsymbol{x}_*, \boldsymbol{x}_i) \right) \\
&= D_h(\boldsymbol{x}_*, \boldsymbol{x}_0) - D_h(\boldsymbol{x}_*, \boldsymbol{x}_k) \\
&\leq D_h(\boldsymbol{x}_*, \boldsymbol{x}_0).
\end{aligned}
$$

Hence, by dividing the both sides of the inequality by $(1 - \rho L_h)k$, we get

$$
\min_{i=1,\dots,k} \min_{\boldsymbol{s}_i \in \boldsymbol{Mx}_i} \frac{\|\boldsymbol{s}_i\|^2}{2L_h} \leq \min_{i=1,\dots,k} D_h(\boldsymbol{x}_i, \boldsymbol{x}_{i-1}) \leq \frac{1}{k} \sum_{i=1}^{k} D_h(\boldsymbol{x}_i, \boldsymbol{x}_{i-1}) \leq \frac{D_h(\boldsymbol{x}_*, \boldsymbol{x}_0)}{(1 - \rho L_h)k}.
$$

The first inequality follows from $\nabla h(\boldsymbol{x}_i) - \nabla h(\boldsymbol{x}_{i-1}) \in \boldsymbol{Mx}_i$, and the convex and $L_h$-smooth properties of $h$, *i.e.*, $\frac{1}{2L_h}\|\nabla h(\boldsymbol{x}) - \nabla h(\boldsymbol{y})\|^2 \leq D_h(\boldsymbol{x}, \boldsymbol{y})$ (Nesterov, 2018, Theroem 2.1.5).

Next, we prove that all limit point of the sequence $\{\boldsymbol{x}_k\}$ are first-order stationary points. By (8), the sequence $\{D_h(\boldsymbol{x}_*, \boldsymbol{x}_k)\}$ is bounded above, which implies that $\{\boldsymbol{x}_k\}$ is a bounded sequence due to the strong convexity of $h$. Let $\boldsymbol{x}_\infty$ be any limit point of $\{\boldsymbol{x}_k\}$, and let $\{\boldsymbol{x}_{k(j)}\}$ be a subsequence that converges to $\boldsymbol{x}_\infty$. Note that $D_h(\boldsymbol{x}_{k+1}, \boldsymbol{x}_k) \to 0$ (and $\nabla h(\boldsymbol{x}_k) - \nabla h(\boldsymbol{x}_{k+1}) \to \boldsymbol{0}$ by the convex and $L_h$-smooth properties of $h$, *i.e.*, $\frac{1}{2L_h}\|\nabla h(\boldsymbol{x}) - \nabla h(\boldsymbol{y})\|^2 \leq D_h(\boldsymbol{x}, \boldsymbol{y})$ (Nesterov, 2018, Theroem 2.1.5)) as $k \to \infty$, since $\sum_{i=1}^{k} D_h(\boldsymbol{x}_i, \boldsymbol{x}_{i-1}) \leq \frac{D_h(\boldsymbol{x}_*, \boldsymbol{x}_0)}{1 - \rho L_h}$. Then $\{\boldsymbol{x}_{k(j)+1}\}$ also converges to $\boldsymbol{x}_\infty$ since

$$
\begin{aligned}
\|\boldsymbol{x}_{k(j)+1} - \boldsymbol{x}_\infty\|^2 &\leq 2\|\boldsymbol{x}_{k(j)} - \boldsymbol{x}_\infty\|^2 + 2\|\boldsymbol{x}_{k(j)+1} - \boldsymbol{x}_{k(j)}\|^2 \\
&\leq 2\|\boldsymbol{x}_{k(j)} - \boldsymbol{x}_\infty\|^2 + \frac{4}{\mu_h} D_h(\boldsymbol{x}_{k(j)+1}, \boldsymbol{x}_{k(j)}),
\end{aligned}
$$

where the last inequality uses the strong convexity of $h$, *i.e.*, $\frac{\mu_h}{2}\|\boldsymbol{x}_{k(j)+1} - \boldsymbol{x}_{k(j)}\|^2 \leq D_h(\boldsymbol{x}_{k(j)+1}, \boldsymbol{x}_{k(j)})$. Since $\boldsymbol{M} + \gamma \boldsymbol{I}$ is maximally monotone and satisfies $\nabla h(\boldsymbol{x}_{k(j)}) - \nabla h(\boldsymbol{x}_{k(j)+1}) + \gamma \boldsymbol{x}_{k(j)+1} \in (\boldsymbol{M} + \gamma \boldsymbol{I})(\boldsymbol{x}_{k(j)+1})$, we finally have $\boldsymbol{0} \in \boldsymbol{Mx}_\infty$ by Bauschke & Combettes (2011, Proposition 20.32).

Lastly, assume that $X_*^\rho(\boldsymbol{M}) = X_*(\boldsymbol{M})$. Then $\boldsymbol{x}_\infty$ is in $X_*^\rho(\boldsymbol{M})$, and since $\lim_{j\to\infty} D_h(\boldsymbol{x}_\infty, \boldsymbol{x}_{k(j)}) = 0$ and $\{D_h(\boldsymbol{x}_\infty, \boldsymbol{x}_k)\}$ is a nonincreasing sequence by (8), we get $\lim_{k\to\infty} D_h(\boldsymbol{x}_\infty, \boldsymbol{x}_k) = 0$. Therefore, by the strong convexity of $h$, $\{\boldsymbol{x}_k\}$ converges to $\boldsymbol{x}_\infty \in X_*^\rho(\boldsymbol{M})$.

## C.4 SEPARATING HYPERPLANE

We prove that $H(\boldsymbol{x})$ separates $\boldsymbol{x}$ and $X_*^\rho(\boldsymbol{M})$ by showing the following two inequalities:

$$\langle \nabla h(\boldsymbol{x}) - \nabla h(\boldsymbol{R}\boldsymbol{x}), \, \boldsymbol{x} - \boldsymbol{x} \rangle < \left( \frac{1}{L_h} - \frac{\rho}{2} \right) ||\nabla h(\boldsymbol{x}) - \nabla h(\boldsymbol{R}\boldsymbol{x})||^2,$$

$$\langle \nabla h(\boldsymbol{x}) - \nabla h(\boldsymbol{R}\boldsymbol{x}), \, \boldsymbol{x} - \boldsymbol{x}_* \rangle \geq \left( \frac{1}{L_h} - \frac{\rho}{2} \right) ||\nabla h(\boldsymbol{x}) - \nabla h(\boldsymbol{R}\boldsymbol{x})||^2,$$

for all $\boldsymbol{x}_* \in X_*^\rho(\boldsymbol{M})$.

The first inequality is straightforward as the left-hand side of the inequality is equal to zero and the right-hand side of the inequality is positive (unless $\nabla h(\boldsymbol{x}) = \nabla h(\boldsymbol{R}\boldsymbol{x})$) under the condition $L_h \rho < 2$.

Next, for any solution $\boldsymbol{x}_* \in X_*^\rho(\boldsymbol{M})$, we get

$$\langle \nabla h(\boldsymbol{x}) - \nabla h(\boldsymbol{R}\boldsymbol{x}), \, \boldsymbol{x} - \boldsymbol{x}_* \rangle = \langle \nabla h(\boldsymbol{x}) - \nabla h(\boldsymbol{R}\boldsymbol{x}), \, (\boldsymbol{x} - \boldsymbol{R}\boldsymbol{x}) + (\boldsymbol{R}\boldsymbol{x} - \boldsymbol{x}_*) \rangle$$

$$\geq \left( \frac{1}{L_h} - \frac{\rho}{2} \right) ||\nabla h(\boldsymbol{x}) - \nabla h(\boldsymbol{R}\boldsymbol{x})||^2, \qquad (9)$$

which uses the $\frac{1}{L_h}$-cocoercivity of $\nabla h$, i.e., $\frac{1}{L_h} ||\nabla h(\boldsymbol{x}) - \nabla h(\boldsymbol{y})||^2 \leq \langle \nabla h(\boldsymbol{x}) - \nabla h(\boldsymbol{y}), \, \boldsymbol{x} - \boldsymbol{y} \rangle$ (Nesterov, 2018, Theroem 2.1.5), and the weak MVI condition (Assumption A3) with $\nabla h(\boldsymbol{x}) - \nabla h(\boldsymbol{R}\boldsymbol{x}) \in \boldsymbol{M}\boldsymbol{R}\boldsymbol{x}$.

## C.5 PROOF OF LEMMA 3

By the definition $\tilde{\boldsymbol{R}}\boldsymbol{x} := \boldsymbol{I} - \left( \frac{1}{L_h} - \frac{\rho}{2} \right)(\nabla h - \nabla h \boldsymbol{R})$, we get

$$||\tilde{\boldsymbol{R}}\boldsymbol{x} - \boldsymbol{x}_*||^2 = ||\boldsymbol{x} - \boldsymbol{x}_*||^2 - 2\left( \frac{1}{L_h} - \frac{\rho}{2} \right) \langle \nabla h(\boldsymbol{x}) - \nabla h(\boldsymbol{R}\boldsymbol{x}), \, \boldsymbol{x} - \boldsymbol{x}_* \rangle$$

$$+ \left( \frac{1}{L_h} - \frac{\rho}{2} \right)^2 ||\nabla h(\boldsymbol{x}) - \nabla h(\boldsymbol{R}\boldsymbol{x})||^2$$

$$\leq ||\boldsymbol{x} - \boldsymbol{x}_*||^2 - \left( \frac{1}{L_h} - \frac{\rho}{2} \right)^2 ||\nabla h(\boldsymbol{x}) - \nabla h(\boldsymbol{R}\boldsymbol{x})||^2$$

$$= ||\boldsymbol{x} - \boldsymbol{x}_*||^2 - ||\tilde{\boldsymbol{R}}\boldsymbol{x} - \boldsymbol{x}||^2,$$

where the inequality uses (9).

## C.6 PROOF OF THEOREM 2

By Lemma 1, the condition $\mu_h > \gamma$ implies that $\boldsymbol{R}\boldsymbol{x}$ exists for any $\boldsymbol{x}$ and thus $\tilde{\boldsymbol{R}}\boldsymbol{x}$ is well-defined. By Lemma 3, we get

$$||\boldsymbol{x}_i - \boldsymbol{x}_*||^2 \leq ||\boldsymbol{x}_{i-1} - \boldsymbol{x}_*||^2 - ||\boldsymbol{x}_i - \boldsymbol{x}_{i-1}||^2 \qquad (10)$$

$$= ||\boldsymbol{x}_{i-1} - \boldsymbol{x}_*||^2 - \left( \frac{1}{L_h} - \frac{\rho}{2} \right)^2 ||\nabla h(\boldsymbol{x}_{i-1}) - \nabla h(\boldsymbol{R}\boldsymbol{x}_{i-1})||^2$$

for all $i \geq 1$. By summing over the above inequality, we get

$$\sum_{i=1}^k \frac{(2 - \rho L_h)^2}{4 L_h^2} ||\nabla h(\boldsymbol{R}\boldsymbol{x}_{i-1}) - \nabla h(\boldsymbol{x}_{i-1})||^2 \leq \sum_{i=1}^k (||\boldsymbol{x}_{i-1} - \boldsymbol{x}_*||^2 - ||\boldsymbol{x}_i - \boldsymbol{x}_*||^2)$$

$$= ||\boldsymbol{x}_0 - \boldsymbol{x}_*||^2 - ||\boldsymbol{x}_k - \boldsymbol{x}_*||^2$$

$$\leq ||\boldsymbol{x}_0 - \boldsymbol{x}_*||^2.$$

Hence, by dividing the both sides of the inequality by $(2 - \rho L_h)^2 k / 2 L_h$, we get

$$\min_{i=1,\dots,k} \frac{||\nabla h(\boldsymbol{R}\boldsymbol{x}_{i-1}) - \nabla h(\boldsymbol{x}_{i-1})||^2}{2 L_h} \leq \frac{1}{k} \sum_{i=1}^k \frac{||\nabla h(\boldsymbol{R}\boldsymbol{x}_{i-1}) - \nabla h(\boldsymbol{x}_{i-1})||^2}{2 L_h} \leq \frac{2 L_h ||\boldsymbol{x}_0 - \boldsymbol{x}_*||^2}{(2 - \rho L_h)^2 k}.$$

By using the fact that $\nabla h(\boldsymbol{Rx}_{i-1}) - \nabla h(\boldsymbol{x}_{i-1}) \in \boldsymbol{MRx}_{i-1}$, the lower bound of the inequality can be further bounded as

$$\min_{i=1,\ldots,k} \min_{\boldsymbol{s}_i \in \boldsymbol{MRx}_{i-1}} \frac{||\boldsymbol{s}_i||^2}{2L_h} \leq \min_{i=1,\ldots,k} \frac{||\nabla h(\boldsymbol{Rx}_{i-1}) - \nabla h(\boldsymbol{x}_{i-1})||^2}{2L_h}.$$

Next, we prove that all limit point of the sequence $\{\boldsymbol{x}_k\}$ are first-order stationary points. By (10), the sequence $\{||\boldsymbol{x}_k - \boldsymbol{x}_*||\}$ is bounded above, which implies that $\{\boldsymbol{x}_k\}$ is a bounded sequence. Let $\boldsymbol{x}_\infty$ be any limit point of $\{\boldsymbol{x}_k\}$, and let $\{\boldsymbol{x}_{k(j)}\}$ be a subsequence that converges to $\boldsymbol{x}_\infty$. Note that $||\nabla h(\boldsymbol{x}_k) - \nabla h(\boldsymbol{Rx}_k)|| \to 0$ (and $||\boldsymbol{x}_k - \boldsymbol{Rx}_k|| \to 0$ by the $\mu_h$-strong convexity of $h$, $i.e.$, $\mu_h||\boldsymbol{x}_k - \boldsymbol{Rx}_k|| \leq ||\nabla h(\boldsymbol{x}_k) - \nabla h(\boldsymbol{Rx}_k)||$) as $k \to \infty$, since $\sum_{i=1}^k \frac{1}{2L_h}||\nabla h(\boldsymbol{x}_{i-1}) - \nabla h(\boldsymbol{Rx}_{i-1})||^2 \leq \frac{2L_h||\boldsymbol{x}_0 - \boldsymbol{x}_*||^2}{(2-\rho L_h)^2}$. Then $\{\boldsymbol{Rx}_{k(j)}\}$ also converges to $\boldsymbol{x}_\infty$ since

$$||\boldsymbol{Rx}_{k(j)} - \boldsymbol{x}_\infty|| \leq ||\boldsymbol{Rx}_{k(j)} - \boldsymbol{x}_{k(j)}|| + ||\boldsymbol{x}_{k(j)} - \boldsymbol{x}_\infty||.$$

Since $\boldsymbol{M} + \gamma\boldsymbol{I}$ is maximally monotone and satisfies $\nabla h(\boldsymbol{x}_{k(j)}) - \nabla h(\boldsymbol{Rx}_{k(j)}) + \gamma\boldsymbol{Rx}_{k(j)} \in (\boldsymbol{M} + \gamma\boldsymbol{I})(\boldsymbol{Rx}_{k(j)})$, we finally have $\boldsymbol{0} \in \boldsymbol{Mx}_\infty$ by Bauschke & Combettes (2011, Proposition 20.32).

Lastly, assume that $X_*^\rho(\boldsymbol{M}) = X_*(\boldsymbol{M})$. Then $\boldsymbol{x}_\infty$ is in $X_*^\rho(\boldsymbol{M})$, and since $\lim_{j\to\infty} ||\boldsymbol{x}_{k(j)} - \boldsymbol{x}_\infty|| = 0$ and $||\boldsymbol{x}_k - \boldsymbol{x}_\infty||$ is a nonincreasing sequence by (10), we get $\lim_{k\to\infty} ||\boldsymbol{x}_k - \boldsymbol{x}_\infty|| = 0$.

# D    Proofs for Section 6

## D.1    Proof of Theorem 5

We first extend Theorem 1 of the (exact) BPP method to its inexact variant that approximately computes the $h$-resolvent in BPP.

**Lemma 4.** *Let $\{\boldsymbol{x}_k\}$ be generated by an inexact BPP, and $\boldsymbol{x}_k^* := \boldsymbol{Rx}_{k-1}$ be an exactly updated point from $\boldsymbol{x}_{k-1}$, where $\boldsymbol{x}_k \neq \boldsymbol{x}_k^*$ in general. Then, under the conditions in Theorem 1, the sequence $\{\boldsymbol{x}_k\}$ satisfies, for $k \geq 1$ and for any $\boldsymbol{x}_* \in X_*^\rho(\boldsymbol{M})$,*

$$\min_{i=1,\ldots,k} D_h(\boldsymbol{x}_i^*, \boldsymbol{x}_{i-1}) \leq \frac{2D_h(\boldsymbol{x}_*, \boldsymbol{x}_0) + \sum_{i=1}^k \frac{i+1}{i}\left((i+1)^2 \frac{L_h}{\mu_h} - 1\right)\frac{L_h}{2}||\boldsymbol{x}_i - \boldsymbol{x}_i^*||^2}{(1-\rho L_h)\sum_{i=1}^k \frac{i+1}{i}}.$$

*Proof.* Since $\nabla h(\boldsymbol{x}_{i-1}) - \nabla h(\boldsymbol{x}_i^*) \in \boldsymbol{Mx}_i^*$, the weak MVI condition implies

$$0 \leq \langle \nabla h(\boldsymbol{x}_{i-1}) - \nabla h(\boldsymbol{x}_i^*), \boldsymbol{x}_i^* - \boldsymbol{x}_* \rangle + \frac{\rho}{2}||\nabla h(\boldsymbol{x}_{i-1}) - \nabla h(\boldsymbol{x}_i^*)||^2$$

$$= D_h(\boldsymbol{x}_*, \boldsymbol{x}_{i-1}) - D_h(\boldsymbol{x}_*, \boldsymbol{x}_i^*) - D_h(\boldsymbol{x}_i^*, \boldsymbol{x}_{i-1}) + \frac{\rho}{2}||\nabla h(\boldsymbol{x}_{i-1}) - \nabla h(\boldsymbol{x}_i^*)||^2$$

$$\leq D_h(\boldsymbol{x}_*, \boldsymbol{x}_{i-1}) - D_h(\boldsymbol{x}_*, \boldsymbol{x}_i^*) - (1-\rho L_h)D_h(\boldsymbol{x}_i^*, \boldsymbol{x}_{i-1})$$

$$= D_h(\boldsymbol{x}_*, \boldsymbol{x}_{i-1}) - D_h(\boldsymbol{x}_*, \boldsymbol{x}_i) + (D_h(\boldsymbol{x}_*, \boldsymbol{x}_i) - D_h(\boldsymbol{x}_*, \boldsymbol{x}_i^*)) - (1-\rho L_h)D_h(\boldsymbol{x}_i^*, \boldsymbol{x}_{i-1}).$$

The term $D_h(\boldsymbol{x}_*, \boldsymbol{x}_i) - D_h(\boldsymbol{x}_*, \boldsymbol{x}_i^*)$ can be further bounded as

$$D_h(\boldsymbol{x}_*, \boldsymbol{x}_i) - D_h(\boldsymbol{x}_*, \boldsymbol{x}_i^*) = h(\boldsymbol{x}_i^*) - h(\boldsymbol{x}_i) - \langle \nabla h(\boldsymbol{x}_i), \boldsymbol{x}_* - \boldsymbol{x}_i \rangle + \langle \nabla h(\boldsymbol{x}_i^*), \boldsymbol{x}_* - \boldsymbol{x}_i^* \rangle$$

$$= -D_h(\boldsymbol{x}_i, \boldsymbol{x}_i^*) + \langle \nabla h(\boldsymbol{x}_i^*) - \nabla h(\boldsymbol{x}_i), \boldsymbol{x}_* - \boldsymbol{x}_i \rangle$$

$$\leq -D_h(\boldsymbol{x}_i, \boldsymbol{x}_i^*) + \frac{\eta}{2\mu_h}||\nabla h(\boldsymbol{x}_i^*) - \nabla h(\boldsymbol{x}_i)||^2 + \frac{\mu}{2\eta}||\boldsymbol{x}_* - \boldsymbol{x}_i||^2$$

$$\leq -D_h(\boldsymbol{x}_i, \boldsymbol{x}_i^*) + \eta\frac{L_h}{\mu_h}D_h(\boldsymbol{x}_i, \boldsymbol{x}_i^*) + \frac{1}{\eta}D_h(\boldsymbol{x}_*, \boldsymbol{x}_i)$$

$$= \left(\eta\frac{L_h}{\mu_h} - 1\right)D_h(\boldsymbol{x}_i, \boldsymbol{x}_i^*) + \frac{1}{\eta}D_h(\boldsymbol{x}_*, \boldsymbol{x}_i). \tag{11}$$

Therefore, we get

$$(1-\rho L_h)D_h(\boldsymbol{x}_i^*, \boldsymbol{x}_{i-1}) \leq D_h(\boldsymbol{x}_*, \boldsymbol{x}_{i-1}) - \left(1 - \frac{1}{\eta}\right)D_h(\boldsymbol{x}_*, \boldsymbol{x}_i) + \left(\eta\frac{L_h}{\mu_h} - 1\right)D_h(\boldsymbol{x}_i, \boldsymbol{x}_i^*),$$

and by taking $\eta = (i+1)^2$ and multiplying both sides by $\frac{i+1}{i}$, we get

$$
\begin{aligned}
\frac{i+1}{i}(1 - \rho L_h)D_h(\boldsymbol{x}_i^*, \boldsymbol{x}_{i-1}) \leq & \frac{i+1}{i}D_h(\boldsymbol{x}_*, \boldsymbol{x}_{i-1}) - \frac{i+2}{i+1}D_h(\boldsymbol{x}_*, \boldsymbol{x}_i) \\
& + \frac{i+1}{i}\left((i+1)^2\frac{L_h}{\mu_h} - 1\right)D_h(\boldsymbol{x}_i, \boldsymbol{x}_i^*) \\
\leq & \frac{i+1}{i}D_h(\boldsymbol{x}_*, \boldsymbol{x}_{i-1}) - \frac{i+2}{i+1}D_h(\boldsymbol{x}_*, \boldsymbol{x}_i) \\
& + \frac{i+1}{i}\left((i+1)^2\frac{L_h}{\mu_h} - 1\right)\frac{L_h}{2}||\boldsymbol{x}_i - \boldsymbol{x}_i^*||^2. \quad (12)
\end{aligned}
$$

Then the result follows directly by summing over the inequalities for all $i = 1, \dots, k$ and dividing both sides by $(1 - \rho L_h)\sum_{i=1}^k \frac{i+1}{i}$. $\qquad\square$

This leads to the following theorem, and consequently we have Theorem 5. Here, the choice of $J$ depends on the total number of outer iterations $k$, so this will be mostly useful when we are given $k$ in advance. We omitted here, but it is possible to modify the proof to have same complexity result with varying number of inner iterations $J(i)$ that increases in the order of $O(\log(i))$.

**Theorem 7.** *Let $M$ (2) of the composite problem (1) satisfy Assumption A3 for some $\rho \in \left[0, \frac{1}{2L+\tilde{L}}\right)$, and let $f, g$ and $\phi$ satisfy Assumption A1 Then, the sequence $\{\boldsymbol{x}_k\}$ of the SA-MGDA (with a finite $J$) satisfies, for $k \geq 1$, $\tau \in \left(\frac{\rho}{1-\rho L}, \frac{1}{L+\tilde{L}}\right)$ and for any $\boldsymbol{x}_* \in X_*^\rho(M)$,*

$$
\min_{i=1,\dots,k} \min_{\boldsymbol{s}_i \in \boldsymbol{MRx}_{i-1}} \frac{||\boldsymbol{s}_i||^2}{2\left(\frac{1}{\tau}+L\right)} \leq \min_{i=1,\dots,k} D_h(\boldsymbol{Rx}_{i-1}, \boldsymbol{x}_{i-1}) \leq \frac{3D_h(\boldsymbol{x}_*, \boldsymbol{x}_0)}{\left(1 - \rho\left(\frac{1}{\tau}+L\right)\right)k},
$$

*for $J \geq \frac{\frac{1}{\tau}+2L_{vv}}{\frac{1}{\tau}-2L_{vv}}\log\left((k+1)(k+3)\left((k+1)^2\frac{1+\tau L}{1-\tau L} - 1\right)\frac{4(1+\tau L)}{1-\tau L}\right)$.*

*Proof.* We first upper bound the term $||\boldsymbol{x}_i - \boldsymbol{x}_i^*||^2$ in the right-hand side of (12). The sequence $\{(\boldsymbol{u}_i, \boldsymbol{v}_i)\}_{i\geq 0}$ of SA-MGDA (for a finite $J$), an instance of the inexact BPP, satisfies $\boldsymbol{u}_i = \boldsymbol{u}_i^*$, so $||\boldsymbol{x}_i^* - \boldsymbol{x}_i|| = ||\boldsymbol{v}_i^* - \boldsymbol{v}_i||$. Thus,

$$
\begin{aligned}
||\boldsymbol{x}_i - \boldsymbol{x}_i^*||^2 &= ||\boldsymbol{v}_i - \boldsymbol{v}_i^*||^2 \\
&\leq ||\boldsymbol{v}_{i-1} - \boldsymbol{v}_i^*||^2 \exp\left(-\frac{\frac{1}{\tau}-2L_{vv}}{\frac{1}{\tau}+2L_{vv}}J\right) \\
&\leq ||\boldsymbol{x}_{i-1} - \boldsymbol{x}_i^*||^2 \exp\left(-\frac{\frac{1}{\tau}-2L_{vv}}{\frac{1}{\tau}+2L_{vv}}J\right) \\
&\leq (2||\boldsymbol{x}_{i-1} - \boldsymbol{x}_*|| + 2||\boldsymbol{x}_i^* - \boldsymbol{x}_*||^2)\exp\left(-\frac{\frac{1}{\tau}-2L_{vv}}{\frac{1}{\tau}+2L_{vv}}J\right) \\
&\leq \left(2||\boldsymbol{x}_{i-1} - \boldsymbol{x}_*|| + \frac{4}{\mu_h}D_h(\boldsymbol{x}_*, \boldsymbol{x}_{i-1})\right)\exp\left(-\frac{\frac{1}{\tau}-2L_{vv}}{\frac{1}{\tau}+2L_{vv}}J\right) \\
&\leq \frac{8}{\mu_h}D_h(\boldsymbol{x}_*, \boldsymbol{x}_{i-1})\exp\left(-\frac{\frac{1}{\tau}-2L_{vv}}{\frac{1}{\tau}+2L_{vv}}J\right) \quad (13)
\end{aligned}
$$

where the second line follows from the fact that $J$ number of (inner) proximal gradient ascent steps satisfy $||\hat{\boldsymbol{v}}_i - \boldsymbol{v}_i^R||^2 \leq ||\boldsymbol{v}_{i-1} - \boldsymbol{v}_i^R||^2 \exp\left(-\frac{\frac{1}{\tau}-2L_{vv}}{\frac{1}{\tau}+2L_{vv}}J\right)$ (by Theorem 10.29 of Beck (2017)), and the fifth line uses

$$
||\boldsymbol{x}_i^* - \boldsymbol{x}_*||^2 \leq \frac{2}{\mu_h}D_h(\boldsymbol{x}_*, \boldsymbol{x}_i^*) \leq \frac{2}{\mu_h}D_h(\boldsymbol{x}_*, \boldsymbol{x}_{i-1})
$$

due to the strong convexity and the quasi-Bregman nonexpansivity. Then, we have the following upper bound of the right-hand side of the inequality (12)

$$
\begin{aligned}
\frac{i+1}{i}(1-\rho L_h)D_h(\boldsymbol{x}_i^*, \boldsymbol{x}_{i-1}) \leq & \frac{i+1}{i}D_h(\boldsymbol{x}_*, \boldsymbol{x}_{i-1}) - \frac{i+2}{i+1}D_h(\boldsymbol{x}_*, \boldsymbol{x}_i) \\
& + \frac{i+1}{i}\left((i+1)^2\frac{L_h}{\mu_h} - 1\right)\frac{4L_h}{\mu_h}D_h(\boldsymbol{x}_*, \boldsymbol{x}_{i-1})\exp\left(-\frac{\frac{1}{\tau}-2L_{\boldsymbol{vv}}}{\frac{1}{\tau}+2L_{\boldsymbol{vv}}}J\right) \\
\leq & \left(1+\frac{1}{(i+1)(i+3)}\right)\frac{i+1}{i}D_h(\boldsymbol{x}_*, \boldsymbol{x}_{i-1}) - \frac{i+2}{i+1}D_h(\boldsymbol{x}_*, \boldsymbol{x}_i),
\end{aligned}
$$

where the second inequality uses $J \geq \frac{\frac{1}{\tau}+2L_{\boldsymbol{vv}}}{\frac{1}{\tau}-2L_{\boldsymbol{vv}}}\log\left((i+1)(i+3)\left((i+1)^2\frac{L_h}{\mu_h}-1\right)\frac{4L_h}{\mu_h}\right)$.

Then, by multiplying $\frac{i+3}{i+2}$ on both sides, we get

$$
\frac{(i+1)(i+3)}{i(i+2)}(1-\rho L_h)D_h(\boldsymbol{x}_i^*, \boldsymbol{x}_{i-1}) \leq \frac{i+2}{i}D_h(\boldsymbol{x}_*, \boldsymbol{x}_{i-1}) - \frac{i+3}{i+1}D_h(\boldsymbol{x}_*, \boldsymbol{x}_i).
$$

Hence, by summing over the inequalities for all $i = 1, \ldots, k$, we get

$$
\sum_{i=1}^{k}\frac{(i+1)(i+3)}{i(i+2)}(1-\rho L_h)D_h(\boldsymbol{x}_i^*, \boldsymbol{x}_{i-1}) \leq 3D_h(\boldsymbol{x}_*, \boldsymbol{x}_0) - \frac{k+3}{k+1}D_h(\boldsymbol{x}_*, \boldsymbol{x}_k),
$$

and further dividing both sides by $\sum_{i=1}^{k}\frac{(i+1)(i+3)}{i(i+2)}(1-\rho L_h)$, we get

$$
\min_{i=1,\ldots,k}\min_{s_i \in \boldsymbol{MRx}_{i-1}}\frac{||s_i||^2}{2\left(\frac{1}{\tau}+L\right)} \leq \min_{i=1,\ldots,k}D_h(\boldsymbol{Rx}_{i-1}, \boldsymbol{x}_{i-1}) \leq \frac{3D_h(\boldsymbol{x}_*, \boldsymbol{x}_0)}{(1-\rho L_h)\sum_{i=1}^{k}\frac{(i+1)(i+3)}{i(i+2)}} \leq \frac{3D_h(\boldsymbol{x}_*, \boldsymbol{x}_0)}{(1-\rho L_h)k}
$$

by using the fact that $\frac{1}{2L_h}||\nabla h(\boldsymbol{Rx}_{i-1}) - \nabla h(\boldsymbol{x}_{i-1})||^2 \leq D_h(\boldsymbol{Rx}_{i-1}, \boldsymbol{x}_{i-1})$ and $\nabla h(\boldsymbol{Rx}_{i-1}) - \nabla h(\boldsymbol{x}_{i-1}) \in \boldsymbol{MRx}_{i-1}$.

The constraints $\mu_h > \hat{L}$ and $\rho L_h < 1$ yields $\tau < \frac{1}{L+\hat{L}}$ and $\tau > \frac{\rho}{1-\rho L}$. We need $\rho < \frac{1}{2L+\hat{L}}$, so that $\tau$ exists.

$\square$

### D.2 PROOF OF THEOREM 6

We first extend Lemma 3 to a version with a new variable $\boldsymbol{x}'$ that will be considered as an approximation to $\tilde{\boldsymbol{R}}\boldsymbol{x}$ in the later analysis. The lemma below reduces to Lemma 3 when $\boldsymbol{x}' = \tilde{\boldsymbol{R}}\boldsymbol{x}$ and $\delta = 0$. Introducing a positive constant $\delta$ here seems redundant, but this becomes useful in the upcoming analysis.

**Lemma 5.** *Under the conditions in Lemma 3, we have*

$$
||\boldsymbol{x}' - \boldsymbol{x}_*||^2 \leq (1+\delta)||\boldsymbol{x} - \boldsymbol{x}_*||^2 - (1+\delta)||\tilde{\boldsymbol{R}}\boldsymbol{x} - \boldsymbol{x}||^2 + \left(1+\frac{1}{\delta}\right)||\tilde{\boldsymbol{R}}\boldsymbol{x} - \boldsymbol{x}'||^2,
$$

*for any $\delta > 0$, $\boldsymbol{x}, \boldsymbol{x}'$, and $\boldsymbol{x}_* \in X_\rho^*(\boldsymbol{M})$.*

*Proof.* We have

$$
\begin{aligned}
||\boldsymbol{x}' - \boldsymbol{x}_*||^2 &= ||\tilde{\boldsymbol{R}}\boldsymbol{x} - \boldsymbol{x}_* + \boldsymbol{x}' - \tilde{\boldsymbol{R}}\boldsymbol{x}||^2 \\
&= ||\tilde{\boldsymbol{R}}\boldsymbol{x} - \boldsymbol{x}_*||^2 + 2\langle \tilde{\boldsymbol{R}}\boldsymbol{x} - \boldsymbol{x}_*, \boldsymbol{x}' - \tilde{\boldsymbol{R}}\boldsymbol{x}\rangle + ||\boldsymbol{x}' - \tilde{\boldsymbol{R}}\boldsymbol{x}||^2 \\
&\leq (1+\delta)||\tilde{\boldsymbol{R}}\boldsymbol{x} - \boldsymbol{x}_*||^2 + \left(1+\frac{1}{\delta}\right)||\boldsymbol{x}' - \tilde{\boldsymbol{R}}\boldsymbol{x}||^2 \\
&\leq (1+\delta)||\boldsymbol{x} - \boldsymbol{x}_*||^2 - (1+\delta)||\tilde{\boldsymbol{R}}\boldsymbol{x} - \boldsymbol{x}||^2 + \left(1+\frac{1}{\delta}\right)||\boldsymbol{x}' - \tilde{\boldsymbol{R}}\boldsymbol{x}||^2,
\end{aligned}
$$

where the first inequality uses Young's inequality, and the second inequality uses Lemma 3. $\square$

Next, using Lemma 5, we extend Theorem 2 of the (exact) BPP with projection method to its inexact variant that approximately computes the $h$-resolvent in the BPP with projection, *i.e.*, $\boldsymbol{x}_{k+1} := \boldsymbol{x}_k - \left(\frac{1}{L_h} - \frac{\rho}{2}\right)(\nabla h(\boldsymbol{x}_k) - \nabla h(\hat{\boldsymbol{x}}_{k+1}))$.

**Lemma 6.** *Let $\{\boldsymbol{x}_k\}$ be generated by an inexact BPP with projection, and $\boldsymbol{x}_k^* := \tilde{\boldsymbol{R}}\boldsymbol{x}_{k-1}$ be an exactly updated point from $\boldsymbol{x}_{k-1}$, where $\boldsymbol{x}_k \neq \boldsymbol{x}_k^*$ in general. Then, under the conditions in Theorem 2, the sequence $\{\boldsymbol{x}_k\}$ satisfies, for $k \geq 1$ and for any $\boldsymbol{x}_* \in X_*^\rho(\boldsymbol{M})$,*

$$\min_{i=1,\ldots,k} \min_{\boldsymbol{s}_i \in \boldsymbol{M}\boldsymbol{R}\boldsymbol{x}_{i-1}} \frac{||\boldsymbol{s}_i||^2}{2L_h} \leq \frac{2L_h\left(2||\boldsymbol{x}_0 - \boldsymbol{x}_*||^2 + \sum_{i=1}^{k}(i+1)(i+2)||\boldsymbol{x}_i - \boldsymbol{x}_i^*||^2\right)}{(2 - \rho L_h)^2 \sum_{i=1}^{k} \frac{i+1}{i}}.$$

*Proof.* By taking $\delta = \frac{1}{(i+1)^2 - 1}$, $\boldsymbol{x} = \boldsymbol{x}_{i-1}$, and $\boldsymbol{x}' = \boldsymbol{x}_i$ in Lemma 5, we get

$$||\boldsymbol{x}_i - \boldsymbol{x}_*||^2 \leq \frac{(i+1)^2}{i(i+2)}||\boldsymbol{x}_{i-1} - \boldsymbol{x}_*||^2 - \frac{(i+1)^2}{i(i+2)}||\boldsymbol{x}_i^* - \boldsymbol{x}_{i-1}||^2 + (i+1)^2||\boldsymbol{x}_i - \boldsymbol{x}_i^*||^2,$$

and by multiplying $\frac{i+2}{i+1}$ both sides and reordering terms, we have

$$\frac{i+1}{i}||\boldsymbol{x}_i^* - \boldsymbol{x}_{i-1}||^2 \leq \frac{i+1}{i}||\boldsymbol{x}_{i-1} - \boldsymbol{x}_*||^2 - \frac{i+2}{i+1}||\boldsymbol{x}_i - \boldsymbol{x}_*||^2 + (i+1)(i+2)||\boldsymbol{x}_i - \boldsymbol{x}_i^*||^2. \tag{14}$$

Hence, by summing over the inequalities for all $i = 1, \ldots, k$, we have

$$\sum_{i=1}^{k} \frac{i+1}{i}||\boldsymbol{x}_i^* - \boldsymbol{x}_{i-1}||^2 \leq 2||\boldsymbol{x}_0 - \boldsymbol{x}_*||^2 - \frac{k+2}{k+1}||\boldsymbol{x}_k - \boldsymbol{x}_*||^2 + \sum_{i=1}^{k}(i+1)(i+2)||\boldsymbol{x}_i - \boldsymbol{x}_i^*||^2.$$

Therefore, by using the fact that $||\boldsymbol{x}_i^* - \boldsymbol{x}_{i-1}||^2 = \frac{(2-\rho L_h)^2}{4L_h^2}||\nabla h(\boldsymbol{R}\boldsymbol{x}_{i-1}) - \nabla h(\boldsymbol{x}_{i-1})||^2$ and dividing both sides by $\frac{(2-\rho L_h)^2}{2L_h} \sum_{i=1}^{k} \frac{i+1}{i}$, we get

$$\min_{i=1,\ldots,k} \frac{||\nabla h(\boldsymbol{R}\boldsymbol{x}_{i-1}) - \nabla h(\boldsymbol{x}_{i-1})||^2}{2L_h} \leq \frac{2L_h\left(2||\boldsymbol{x}_0 - \boldsymbol{x}_*||^2 + \sum_{i=1}^{k}(i+1)(i+2)||\boldsymbol{x}_i - \boldsymbol{x}_i^*||^2\right)}{(2 - \rho L_h)^2 \sum_{i=1}^{k} \frac{i+1}{i}}.$$

By using the fact that $\nabla h(\boldsymbol{R}\boldsymbol{x}_{i-1}) - \nabla h(\boldsymbol{x}_{i-1}) \in \boldsymbol{M}\boldsymbol{R}\boldsymbol{x}_{i-1}$, the left-hand side of the inequality can be further bounded as

$$\min_{i=1,\ldots,k} \min_{\boldsymbol{s}_i \in \boldsymbol{M}\boldsymbol{R}\boldsymbol{x}_{i-1}} \frac{||\boldsymbol{s}_i||^2}{2L_h} \leq \min_{i=1,\ldots,k} \frac{||\nabla h(\boldsymbol{R}\boldsymbol{x}_{i-1}) - \nabla h(\boldsymbol{x}_{i-1})||^2}{2L_h}.$$

$\square$

This leads to the following theorem, and consequently we have Theorem 6. Here, the choice of $J$ depends on the total number of outer iterations $k$, so this will be mostly useful when we are given $k$ in advance. We omitted here, but it is possible to modify the proof to have same complexity result with varying number of inner iterations $J(i)$ that increases in the order of $O(\log(i))$.

**Theorem 8.** *Let $\boldsymbol{M}$ (2) of the composite problem (1) satisfy Assumption A3 for some $\rho \in \left[0, \frac{2}{2L+\tilde{L}}\right)$, and let $f, g$ and $\phi$ satisfy Assumption A1 Then, the sequence $\{\boldsymbol{x}_k\}$ of the SA-MGDA with projection (for a finite $J$) satisfies, for $k \geq 1$, $\tau \in \left(\frac{\rho}{2-\rho L}, \frac{1}{L+\tilde{L}}\right)$ and for any $\boldsymbol{x}_* \in X_*^\rho(\boldsymbol{M})$,*

$$\min_{i=1,\ldots,k} \min_{\boldsymbol{s}_i \in \boldsymbol{M}\boldsymbol{R}\boldsymbol{x}_{i-1}} \frac{||\boldsymbol{s}_i||^2}{2\left(\frac{1}{\tau} + L\right)} \leq \frac{6\left(\frac{1}{\tau} + L\right)||\boldsymbol{x}_0 - \boldsymbol{x}_*||^2}{\left(2 - \rho\left(\frac{1}{\tau} + L\right)\right)^2 k},$$

*for $J \geq \frac{\frac{1}{\tau} + 2L_{\boldsymbol{v}\boldsymbol{v}}}{\frac{1}{\tau} - 2L_{\boldsymbol{v}\boldsymbol{v}}} \log\left((k+1)^2(k+2)(k+3)\left(1 - \frac{\rho\left(\frac{1}{\tau}+L\right)}{2}\right)^2 \frac{4}{1-\tau L}\right)$.*

*Proof.* We first upper bound the term $||\boldsymbol{x}_i - \boldsymbol{x}_i^*||^2$ in the right-hand side of (14). Let us define $(\boldsymbol{u}_i^{\boldsymbol{R}}, \boldsymbol{v}_i^{\boldsymbol{R}}) := \boldsymbol{R}\boldsymbol{x}_{i-1}$.

$$
\begin{aligned}
||\boldsymbol{x}_i - \boldsymbol{x}_i^*||^2 &= \left(\frac{1}{L_h} - \frac{\rho}{2}\right)^2 ||\nabla h(\hat{\boldsymbol{x}}_i) - \nabla h(\boldsymbol{R}\boldsymbol{x}_{i-1})||^2 \\
&\leq \left(1 - \frac{\rho L_h}{2}\right)^2 ||\hat{\boldsymbol{x}}_i - \boldsymbol{R}\boldsymbol{x}_{i-1}||^2 \\
&= \left(1 - \frac{\rho L_h}{2}\right)^2 ||\hat{\boldsymbol{v}}_i - \boldsymbol{v}_i^{\boldsymbol{R}}||^2 \\
&\leq \left(1 - \frac{\rho L_h}{2}\right)^2 ||\boldsymbol{v}_{i-1} - \boldsymbol{v}_i^{\boldsymbol{R}}||^2 \exp\left(-\frac{\frac{1}{\tau} - 2L_{\boldsymbol{vv}}}{\frac{1}{\tau} + 2L_{\boldsymbol{vv}}}J\right) \\
&\leq \left(1 - \frac{\rho L_h}{2}\right)^2 ||\boldsymbol{x}_{i-1} - \boldsymbol{R}\boldsymbol{x}_{i-1}||^2 \exp\left(-\frac{\frac{1}{\tau} - 2L_{\boldsymbol{vv}}}{\frac{1}{\tau} + 2L_{\boldsymbol{vv}}}J\right) \\
&\leq \left(1 - \frac{\rho L_h}{2}\right)^2 \left(2||\boldsymbol{x}_{i-1} - \boldsymbol{x}_*||^2 + 2||\boldsymbol{R}\boldsymbol{x}_{i-1} - \boldsymbol{x}_*||^2\right) \exp\left(-\frac{\frac{1}{\tau} - 2L_{\boldsymbol{vv}}}{\frac{1}{\tau} + 2L_{\boldsymbol{vv}}}J\right) \\
&\leq \left(1 - \frac{\rho L_h}{2}\right)^2 \left(2||\boldsymbol{x}_{i-1} - \boldsymbol{x}_*||^2 + \frac{2L_h}{\mu_h}||\boldsymbol{x}_{i-1} - \boldsymbol{x}_*||^2\right) \exp\left(-\frac{\frac{1}{\tau} - 2L_{\boldsymbol{vv}}}{\frac{1}{\tau} + 2L_{\boldsymbol{vv}}}J\right) \\
&= \left(1 - \frac{\rho L_h}{2}\right)^2 \frac{2(\mu_h + L_h)}{\mu_h}||\boldsymbol{x}_{i-1} - \boldsymbol{x}_*||^2 \exp\left(-\frac{\frac{1}{\tau} - 2L_{\boldsymbol{vv}}}{\frac{1}{\tau} + 2L_{\boldsymbol{vv}}}J\right),
\end{aligned}
$$

where the second line uses the $L_h$-smoothness of $h$, the fourth line follows from the fact that $J$ number of (inner) proximal gradient ascent steps satisfy $||\hat{\boldsymbol{v}}_i - \boldsymbol{v}_i^{\boldsymbol{R}}|| \leq ||\boldsymbol{v}_{i-1} - \boldsymbol{v}_i^{\boldsymbol{R}}|| \exp\left(-\frac{\frac{1}{\tau} - 2L_{\boldsymbol{vv}}}{2(\frac{1}{\tau} + 2L_{\boldsymbol{vv}})}J\right)$ (by Theorem 10.29 of Beck (2017)), and the seventh line uses

$$
||\boldsymbol{R}\boldsymbol{x}_{i-1} - \boldsymbol{x}_*||^2 \leq \frac{2}{\mu_h}D_h(\boldsymbol{x}_*, \boldsymbol{R}\boldsymbol{x}_{i-1}) \leq \frac{2}{\mu_h}D_h(\boldsymbol{x}_*, \boldsymbol{x}_{i-1}) \leq \frac{L_h}{\mu_h}||\boldsymbol{x}_{i-1} - \boldsymbol{x}_*||^2
$$

due to the strong convexity and smoothness of $h$ and the quasi-Bregman nonexpansivity. Then, we have the following upper bound of the right-hand side of the inequality (14)

$$
\begin{aligned}
\frac{i+1}{i}||\boldsymbol{x}_i^* - \boldsymbol{x}_{i-1}||^2 \leq & \frac{i+1}{i}||\boldsymbol{x}_{i-1} - \boldsymbol{x}_*||^2 - \frac{i+2}{i+1}||\boldsymbol{x}_i - \boldsymbol{x}_*||^2 \\
& + (i+1)(i+2)\left(1 - \frac{\rho L_h}{2}\right)^2 \frac{2(\mu_h + L_h)}{\mu_h}||\boldsymbol{x}_{i-1} - \boldsymbol{x}_*||^2 \exp\left(-\frac{\frac{1}{\tau} - 2L_{\boldsymbol{vv}}}{\frac{1}{\tau} + 2L_{\boldsymbol{vv}}}J\right) \\
\leq & \left(1 + \frac{1}{(i+1)(i+3)}\right)\frac{i+1}{i}||\boldsymbol{x}_{i-1} - \boldsymbol{x}_*||^2 - \frac{i+2}{i+1}||\boldsymbol{x}_i - \boldsymbol{x}_*||^2,
\end{aligned}
$$

where the second inequality uses $J \geq \frac{\frac{1}{\tau} + 2L_{\boldsymbol{vv}}}{\frac{1}{\tau} - 2L_{\boldsymbol{vv}}}\log\left(i(i+1)(i+2)(i+3)\left(1 - \frac{\rho L_h}{2}\right)^2 \frac{2(\mu_h + L_h)}{\mu_h}\right)$.
Then, by multiplying $\frac{i+3}{i+2}$ on both sides, we get

$$
\frac{(i+1)(i+3)}{i(i+2)}||\boldsymbol{x}_i^* - \boldsymbol{x}_{i-1}||^2 \leq \frac{i+2}{i}||\boldsymbol{x}_{i-1} - \boldsymbol{x}_*||^2 - \frac{i+3}{i+1}||\boldsymbol{x}_i - \boldsymbol{x}_*||^2.
$$

Hence, by summing over the inequalities for all $i = 1, \ldots, k$, we get

$$
\sum_{i=1}^{k} \frac{(i+1)(i+3)}{i(i+2)}||\boldsymbol{x}_i^* - \boldsymbol{x}_{i-1}||^2 \leq 3||\boldsymbol{x}_0 - \boldsymbol{x}_*||^2 - \frac{k+3}{k+1}||\boldsymbol{x}_k - \boldsymbol{x}_*||^2,
$$

and further dividing both sides by $\sum_{i=1}^{k} \frac{(i+1)(i+3)}{i(i+2)}$, we get

$$
\min_{i=1,\ldots,k} \min_{\boldsymbol{s}_i \in \boldsymbol{M}\boldsymbol{R}\boldsymbol{x}_{i-1}} \frac{(2 - \rho L_h)^2||\boldsymbol{s}_i||^2}{4L_h^2} \leq \min_{i=1,\ldots,k}||\boldsymbol{x}_i^* - \boldsymbol{x}_{i-1}||^2 \leq \frac{3||\boldsymbol{x}_0 - \boldsymbol{x}_*||^2}{\sum_{i=1}^{k} \frac{(i+1)(i+3)}{i(i+2)}} \leq \frac{3||\boldsymbol{x}_0 - \boldsymbol{x}_*||^2}{k}
$$

by using the fact that $||\boldsymbol{x}_i^* - \boldsymbol{x}_{i-1}||^2 = \frac{(2-\rho L_h)^2}{4L_h^2}||\nabla h(\boldsymbol{R}\boldsymbol{x}_{i-1}) - \nabla h(\boldsymbol{x}_{i-1})||^2$ and $\nabla h(\boldsymbol{R}\boldsymbol{x}_{i-1}) - \nabla h(\boldsymbol{x}_{i-1}) \in \boldsymbol{M}\boldsymbol{R}\boldsymbol{x}_{i-1}$.

The constraints $\mu_h > \hat{L}$ and $\rho L_h < 2$ yields $\tau < \frac{1}{L+\hat{L}}$ and $\tau > \frac{\rho}{2-\rho L}$. We need $\rho < \frac{2}{2L+\hat{L}}$, so that $\tau$ exists. $\qquad\square$

# E   BPP AND SA GRADIENT METHODS UNDER STRONG MVI CONDITION

We also consider the strong MVI condition in (Song et al., 2020; Zhou et al., 2017). This condition is also non-monotone, and includes the $\mu$-strong pseudomonotonicity (Nguyen & Qin, 2020) (see (Nguyen & Qin, 2020) for examples). Let $S_*^\mu(\boldsymbol{M})$ be the associated solution set.

**Assumption A4** (Strong MVI). *For some $\mu \geq 0$, there exists a solution $\boldsymbol{x}_* \in X_*(\boldsymbol{M})$ such that*

$$\langle \boldsymbol{x} - \boldsymbol{x}_*, \, \boldsymbol{w} \rangle \geq \mu ||\boldsymbol{x} - \boldsymbol{x}_*||^2, \quad \forall (\boldsymbol{x}, \boldsymbol{w}) \in \operatorname{gra} \boldsymbol{M}.$$

The BPP has a linear rate under the strong MVI condition.

**Theorem 9.** *Let $\boldsymbol{M}$ satisfy Assumptions A2 and A4 for some $\gamma, \mu \geq 0$, and $h$ be a $\mu_h$-strongly convex and $L_h$-smooth Legendre function with $\mu_h > \gamma$. Then, for $k \geq 1$ and for any $\boldsymbol{x}_* \in S_*^\mu(\boldsymbol{M})$, the sequence $\{\boldsymbol{x}_k\}$ of the BPP method (3) satisfies $D_h(\boldsymbol{x}_*, \boldsymbol{x}_k) \leq \left( \frac{2\mu}{L_h} + 1 \right)^{-k} D_h(\boldsymbol{x}_*, \boldsymbol{x}_0)$.*

*Proof.* By Lemma 1, the condition $\mu_h > \gamma$ implies that $\boldsymbol{R}\boldsymbol{x}$ exists for any $\boldsymbol{x}$. By the definition of $\boldsymbol{R}\boldsymbol{x}$, we have $\nabla h(\boldsymbol{x}) - \nabla h(\boldsymbol{R}\boldsymbol{x}) \in \boldsymbol{M}\boldsymbol{R}\boldsymbol{x}$. Then, Assump. A4 on $\boldsymbol{M}$ implies that $\mu||\boldsymbol{x}_* - \boldsymbol{R}\boldsymbol{x}||^2 \leq \langle \nabla h(\boldsymbol{x}) - \nabla h(\boldsymbol{R}\boldsymbol{x}), \, \boldsymbol{R}\boldsymbol{x} - \boldsymbol{x}_* \rangle = -D_h(\boldsymbol{R}\boldsymbol{x}, \boldsymbol{x}) + D_h(\boldsymbol{x}_*, \boldsymbol{x}) - D_h(\boldsymbol{x}_*, \boldsymbol{R}\boldsymbol{x})$. By letting $\boldsymbol{x} = \boldsymbol{x}_{i-1}$ and using the $L_h$-smoothness of $h$, *i.e.*, $D_h(\boldsymbol{x}_*, \boldsymbol{x}_i) \leq \frac{L_h}{2}||\boldsymbol{x}_* - \boldsymbol{x}_i||^2$, we have $\left( \frac{2\mu}{L_h} + 1 \right) D_h(\boldsymbol{x}_*, \boldsymbol{x}_i) \leq -D_h(\boldsymbol{x}_i, \boldsymbol{x}_{i-1}) + D_h(\boldsymbol{x}_*, \boldsymbol{x}_{i-1}) \leq D_h(\boldsymbol{x}_*, \boldsymbol{x}_{i-1})$. $\qquad\square$

The following theorem of the SA-GDmax method is a byproduct of Lemma 1 and Theorem 9 of the BPP method, for a specific $h$ in (5) that is $\mu_h$-strongly convex and $L_h$-smooth with $\mu_h = \frac{1}{\tau} - L$ and $L_h = \frac{1}{\tau} + L$.

**Theorem 10.** *Let $\boldsymbol{M}$ (2) of the composite problem (1) satisfy Assump. A4 for $\mu \geq 0$, and let $f, g$ and $\phi$ satisfy Assump. A1. Then, for $k \geq 1$, $\tau \in \left( 0, \frac{1}{L+\hat{L}} \right)$ and for any $\boldsymbol{x}_* \in S_*^\mu(\boldsymbol{M})$, the sequence of the SA-GDmax satisfies $D_h(\boldsymbol{x}_*, \boldsymbol{x}_k) \leq \left( 1 + \frac{2\tau\mu}{1+\tau L} \right)^{-k} D_h(\boldsymbol{x}_*, \boldsymbol{x}_0)$.*

*Proof.* The proof follows from Theorem 9 with $\mu_h > \hat{L}$. $\qquad\square$

We next analyze the SA-MGDA under the strong MVI condition. Note that the following theorem reduces to Theorem 10 as $J \to \infty$.

**Theorem 11.** *Let $\boldsymbol{M}$ (2) of the composite problem (1) satisfy Assump. A4 for $\mu \geq 0$, and let $f, g$ and $\phi$ satisfy Assump. A1. Then, for $k \geq 1$, $\tau \in \left( 0, \frac{1}{L+\hat{L}} \right)$ and for any $\boldsymbol{x}_* \in S_*^\mu(\boldsymbol{M})$, the sequence of the SA-MGDA satisfies*

$$D_h(\boldsymbol{x}_*, \boldsymbol{x}_k) \leq \left( \left( \frac{1}{2} + \frac{\tau\mu}{1+\tau L} \right)^{-1} + \frac{8(1+3\tau L)(1+\tau L)}{(1-\tau L)^2} \exp\left( -\frac{\frac{1}{\tau} - 2L_{\boldsymbol{v}\boldsymbol{v}}}{\frac{1}{\tau} + 2L_{\boldsymbol{v}\boldsymbol{v}}} J \right) \right)^k D_h(\boldsymbol{x}_*, \boldsymbol{x}_0).$$

For the proof of Theorem 11, we extend Theorem 9 of the (exact) BPP method to its inexact variant below.

**Lemma 7.** *Let $\{\boldsymbol{x}_k\}$ be generated by an inexact BPP, and $\boldsymbol{x}_k^* := \boldsymbol{R}\boldsymbol{x}_{k-1}$ be an exactly updated point from $\boldsymbol{x}_{k-1}$, where $\boldsymbol{x}_k \neq \boldsymbol{x}_k^*$ in general. Then, under the conditions in Theorem 9, the sequence*

$\{\boldsymbol{x}_k\}$ *satisfies, for* $k \geq 1$ *and for any* $\boldsymbol{x}_* \in S_*^\mu(\boldsymbol{M})$,

$$D_h(\boldsymbol{x}_*, \boldsymbol{x}_k) \leq \left(\frac{\mu}{L_h} + \frac{1}{2}\right)^{-k} D_h(\boldsymbol{x}_*, \boldsymbol{x}_0)$$
$$+ \sum_{i=1}^{k} \left(\frac{\mu}{L_h} + \frac{1}{2}\right)^{-i+1} \left(\frac{2L_h}{\mu_h} - 1\right) L_h ||\boldsymbol{x}_i - \boldsymbol{x}_i^*||^2.$$

*Proof.* Since $\nabla h(\boldsymbol{x}_{i-1}) - \nabla h(\boldsymbol{x}_i^*) \in \boldsymbol{M}\boldsymbol{x}_i^*$, the strong MVI condition implies

$$\mu ||\boldsymbol{x}_* - \boldsymbol{x}_i^*||^2 \leq \langle \nabla h(\boldsymbol{x}_{i-1}) - \nabla h(\boldsymbol{x}_i^*), \boldsymbol{x}_i^* - \boldsymbol{x}_* \rangle$$
$$= D_h(\boldsymbol{x}_*, \boldsymbol{x}_{i-1}) - D_h(\boldsymbol{x}_*, \boldsymbol{x}_i^*) - D_h(\boldsymbol{x}_i^*, \boldsymbol{x}_{i-1}).$$

Since $D_h(\boldsymbol{x}_*, \boldsymbol{x}_i^*) \leq \frac{L_h}{2} ||\boldsymbol{x}_* - \boldsymbol{x}_i^*||^2$, we have

$$\left(\frac{2\mu}{L_h} + 1\right) D_h(\boldsymbol{x}_*, \boldsymbol{x}_i^*) \leq D_h(\boldsymbol{x}_*, \boldsymbol{x}_{i-1}) - D_h(\boldsymbol{x}_i^*, \boldsymbol{x}_{i-1}) \leq D_h(\boldsymbol{x}_*, \boldsymbol{x}_{i-1}).$$

Therefore,

$$D_h(\boldsymbol{x}_*, \boldsymbol{x}_i) \leq \left(\frac{2\mu}{L_h} + 1\right)^{-1} D_h(\boldsymbol{x}_*, \boldsymbol{x}_{i-1}) + (D_h(\boldsymbol{x}_*, \boldsymbol{x}_i) - D_h(\boldsymbol{x}_*, \boldsymbol{x}_i^*))$$
$$\leq \left(\frac{2\mu}{L_h} + 1\right)^{-1} D_h(\boldsymbol{x}_*, \boldsymbol{x}_{i-1}) + \left(\eta\frac{L_h}{\mu_h} - 1\right) D_h(\boldsymbol{x}_i, \boldsymbol{x}_i^*) + \frac{1}{\eta} D_h(\boldsymbol{x}_*, \boldsymbol{x}_i)$$
$$\leq \left(\frac{2\mu}{L_h} + 1\right)^{-1} D_h(\boldsymbol{x}_*, \boldsymbol{x}_{i-1}) + \left(\eta\frac{L_h}{\mu_h} - 1\right) \frac{L_h}{2} ||\boldsymbol{x}_i - \boldsymbol{x}_i^*||^2 + \frac{1}{\eta} D_h(\boldsymbol{x}_*, \boldsymbol{x}_i).$$

where the second inequality follows from (11). Then by subtracting $\frac{1}{\eta} D_h(\boldsymbol{x}_*, \boldsymbol{x}_i)$ and dividing $\left(1 - \frac{1}{\eta}\right)$ both sides, we get

$$D_h(\boldsymbol{x}_*, \boldsymbol{x}_i) \leq \left(1 - \frac{1}{\eta}\right)^{-1} \left(\frac{2\mu}{L_h} + 1\right)^{-1} D_h(\boldsymbol{x}_*, \boldsymbol{x}_{i-1}) + \left(1 - \frac{1}{\eta}\right)^{-1} \left(\eta\frac{L_h}{\mu_h} - 1\right) \frac{L_h}{2} ||\boldsymbol{x}_i - \boldsymbol{x}_i^*||^2.$$
(15)

Then the result follows directly by taking $\eta = 2$ and recursively applying the inequalities for all $i = 1, \ldots, k$. $\qquad\square$

Then, similar to the proof of Theorem 5 in Appendix D.1, we get

$$D_h(\boldsymbol{x}_*, \boldsymbol{x}_i) \leq \left(\frac{\mu}{L_h} + \frac{1}{2}\right)^{-1} D_h(\boldsymbol{x}_*, \boldsymbol{x}_{i-1}) + \left(\frac{2L_h}{\mu_h} - 1\right) L_h ||\boldsymbol{x}_i - \boldsymbol{x}_i^*||^2$$
$$\leq \left(\frac{\mu}{L_h} + \frac{1}{2}\right)^{-1} D_h(\boldsymbol{x}_*, \boldsymbol{x}_{i-1}) + \left(\frac{2L_h}{\mu_h} - 1\right) \frac{8L_h}{\mu_h} D_h(\boldsymbol{x}_*, \boldsymbol{x}_{i-1}) \exp\left(-\frac{\frac{1}{\tau} - 2L_{vv}}{\frac{1}{\tau} + 2L_{vv}} J\right)$$
$$= \left(\left(\frac{\mu}{L_h} + \frac{1}{2}\right)^{-1} + \left(\frac{2L_h}{\mu_h} - 1\right) \frac{8L_h}{\mu_h} \exp\left(-\frac{\frac{1}{\tau} - 2L_{vv}}{\frac{1}{\tau} + 2L_{vv}} J\right)\right) D_h(\boldsymbol{x}_*, \boldsymbol{x}_{i-1}),$$

where the first inequality is (15) by taking $\eta = 2$. Then the result follows directly by recursively applying the inequalities for all $i = 1, \ldots, k$. By Theorem 11, we have the following corollary.

**Corollary 1.** *Under the conditions in Theorem 11, the SA-MGDA method achieves* $D_h(\boldsymbol{x}_*, \boldsymbol{x}_k) \leq \epsilon$ *with* $k = O(\log(\epsilon^{-1}))$ *number of outer iterations and* $J = O(1)$ *number of inner iterations, requiring total* $O(\log(\epsilon^{-1}))$ *gradient computations.*

# F   DETAILS OF THE STRUCTURE OF THE NEURAL NETWORK

| Layer Type | Shape |
|---|---|
| Convolution + tanh | $3 \times 3 \times 5$ |
| Max Pooling | $2 \times 2$ |
| Convolution | $3 \times 3 \times 10$ |
| Max Pooling | $2 \times 2$ |
| Fully Connected + tanh | 250 |
| Fully Connected + tanh | 100 |
| Softmax | 3 |

Table 2: Details of the Structure of the Neural Network

