# OpenReview forum: "Semi-Anchored Gradient Methods for Nonconvex-Nonconcave Minimax Problems"
_ICLR.cc/2024/Conference — Submitted to ICLR 2024_

### Official Review · Reviewer_kCcC · 2023-10-27

**Soundness:** 3 good
**Presentation:** 2 fair
**Contribution:** 2 fair
**Rating:** 3
**Confidence:** 3

**Summary:**

This paper focuses on nonconvex-nonconcave minimax problems. It introduces a new method called the semi-anchored (SA) gradient method, which extend the idea of PDHG to the nonlinear setting by incorperating the certain Bregman distance as a preconditioner. With a designed Legendre function, the SA-GDmax and its practical version SA-MGDA are studied with convergence result and suitable optimality measure.

**Strengths:**

The convergence theorem of proposed algorithm is presented, along with an inexact practical version. Numerical results validate the effectiveness of the proposed algorithm in solving problems that satisfy the weak MVI condition, showing performance comparable to extragradient-type algorithms.

**Weaknesses:**

The paper's motivation should be elucidated in greater detail. Additionally, it is advisable to compare the proposed algorithm with recent papers on nonconvex-nonconcave minimax problems that are based on various regularity conditions, such as dominant conditions and the PL inequality, in order to demonstrate the competitiveness of the proposed approach.

**Questions:**

1. The paper is centered on the one-sided extrapolation-based PDHG method, and while all theoretical performance are similar to the extragradient method under weak MVI conditions, the motivation behind introducing this method may benefit from further clarification. The author also alludes to the potential for improving the extragradient method; providing more specific details on such improvements would enhance the paper.
2. In Theorems 5 and 6, the use of gradient computational cost may not be ideal. The $\mathcal{O}(\log(1/\epsilon))$ cost pertains to the iteration cost of the proximal gradient descent method, where the computational cost of the proximal operator is neglected. Additionally,  the worst computational cost for this class of functions can be  $\mathcal{O}(1/\epsilon)$ as mentioned in arxiv:2101.11041.
3. What is the numerical performance of the algorithm for nonconvex-nonconcave problems without the weak MVI condition? If it performs well in such cases, it may be worth exploring the possibility of relaxing certain conditions to accommodate a broader range of problems. Verifying weak MVI conditions can be challenging, and the need for each stationary point to meet this requirement in the derived theorems could be a limiting factor.

---

> ### Author Response · Authors · 2023-11-18
>
> We appreciate the reviewer's nice summary of our paper, and constructive feedback.
> - (Q1) Regarding the clarification of our contributions, see our general comment.
> - (Q2) You are correct that we omitted the computational cost of the proximal operator. We clarified in revision that we are considering the prox-friendly function, which makes the computational cost of the proximal operation negligible.
> - (Q3) The fair classification experiment is the problem possibly without the weak MVI condition, which is exactly the setting the reviewer is asking for. We clarified this in the paper. / We agree with the reviewer's view on the weak MVI condition, and we are also interested in further relaxing the weak MVI condition, but this does not seem easy at the moment, even for the extragradient. / Regarding the dominant and PL conditions mentioned in the weaknesses section, the fair classification problem is linear in the max-player, so it is obvious that the dominant and PL conditions do not hold. (The toy example also does not satisfy those conditions.) Nevertheless, we have already ran the GDmax (without regularization) that is shown to work under the PL condition in the max-player (but not under the weak MVI), which was slower than the SA-GDmax in our fair classification experiment.

---

### Official Review · Reviewer_Mhw9 · 2023-10-30

**Soundness:** 3 good
**Presentation:** 3 good
**Contribution:** 2 fair
**Rating:** 6
**Confidence:** 4

**Summary:**

The paper consider Bregman divergence based methods for weak Minty variational inequalities. They show convergence of the Bregman divergence between two consecutive iterates for the Bregman proximal point (BPP) method by expressing the scheme as a preconditioned resolvent. By applying the hyperplane projection step of Solodov & Svaiter 1999 they increase the range of $\rho$ (showing convergence in terms of the tangent residual). By modifying the preconditioner further they obtain a scheme which alternatingly computes a proximal gradient for the min-player and solves a proximal (implicit) update for the max-player. They immediately obtain convergence from the previous results. Finally they consider inexactness of the max-player for which they show convergence for the tangent residual.

**Strengths:**

The paper is easy to follow, provides an in depth overview of the relevant literature, and the statements appears correct.

**Weaknesses:**

My main concern is with the relevance of the results:

The only result that seems to exploit the Bregman divergence is regarding the (implicit) BPP without hyperplane projection (Thm. 1 and Thm. 3), and this follows almost immediately from the monotone case.

All remaining results in the paper instead shows rates for the _tangent residual_ (as soon as either inexactness appears or the hyperplane projection is used). If we are interested in the tangent residual in the first place, then a  $\mathcal O(1/k)$ rate can be achieved by an _explicit_ scheme  _without_ (inexact) max-oracles by a primal-dual extragradient scheme.

Consider for instance Algorithm 3 and the associated Theorem 8.2 of [Pethick et al. 2023](https://openreview.net/pdf?id=ejR4E1jaH9k). Without stochasticity a $\mathcal O(1/k)$ rate is recovered for the tangent residual (so no log factor as is otherwise the case when using inexact max-oracles). Algorithm 3 could be simplified further by observing that the bias correction term can be ignored in the deterministic case.

I am left wondering:

- what is the purpose of considering hyperplane projection and inexactness if we cannot provide guarantees in terms of Bregman divergence?
- why not consider the setting without Bregman, and study a nonlinear variant of PDHG (which seemed to be the original motivation in the abstract and which variants for the convex-concave case is mentioned on page 4)? This would essential be an optimistic variant of the PDEG scheme mentioned above.

**Questions:**

- I'm surprised that bounded domain is needed in Thm. 5. Is it not possible to use that inexact proximal point is (approximately) nonexpansive up to an error you can control (through the approximate subsolver of the max-oracle)?

---

> ### Author Response · Authors · 2023-11-18
>
> We appreciate the reviewer's nice summary of our paper, and constructive feedback, especially on the bounded domain assumption.
> - (W1) Regarding the clarification of our contributions in terms of the Bregman distance, see our general comment.
> - (W2) We agree that it would have been the best if we were able to come up with an explicit method that exactly reduces to the PDHG for the bilinear problem and that is comparable to the extragradient in terms of the convergence rate under the weak MVI. This, however, was not a simple task, which we hoped to resolve by starting from the Bregman proximal point perspective. Nevertheless, we still believe our finding has a merit, and we are sharing this work in the hope that this becomes a foundation of developing a decent PDHG-type method for general minimax problems.
> - (Q1) We really appreciate your feedback, and as you expected, we were able to simply remove the bounded domain condition in Theorem 5 of the SA-MGDA, by adopting the flow of the proof of Theorem 6.

---

> > ### Comment · Reviewer_Mhw9 · 2023-11-22
> >
> > I thank the authors for their response, but maintain my position concerning the weaknesses. Specifically:
> >
> > - I think you need to state in Table 1 that a primal-dual extragradient can achieve O(1/k) rate for the squared tangent residual even for weak MVIs (so no need for a max-oracle). In that light, the contribution reduces to extending to Bregman divergences (by sacrificing a log factor).
> > - I only noticed on a second read that the Bregman generator $h$ is required to be both strongly-convex and Lipschitz, which leaves out almost all interesting cases except for the euclidean case. Because of this is seems that even the extension to Bregman is vacuous.
> >
> > On the technical side:
> >
> > - where do you prove that your update BPP is well-defined for weak MVI (A.3)? (It follows in the cohypomonotone case by Bauschke et al. 2019, but otherwise?)

---

> ### Author Response · Authors · 2023-11-22
>
> We are grateful to the reviewer for the prompt response. We have addressed the concern raised by the reviewer below.
>
> - As suggested, we will add the PDEG (next to EG+/CEG+) in Table 1.However, we would like to note that adding the PDEG does not weaken our contribution (which we clarified in our revision), as we already have mentioned EG+ and CEG+ in Table 1 that have the same $O(1/k)$ rate for the squared tangent residual and the computational complexity that is equivalent to the PDEG. We would like to also mention that PDEG is a Gauss-Seidel version of EG, and does not resemble PDHG, although the name looks similar.
> - The strong convexity and Lipschitzness conditions are indeed strong and satisfies only by the Euclidean distance among the existing list of Legendre functions $h$ (some of them are given in the bottom of page 2). We would like to emphasize again that our main contribution is to consider a new (strongly convex and smooth) Legendre function $h$ in (5) that is specifically designed for minimax problems, inspired by the PDHG. This led to the development of the SA gradient method.
> - Lemma 1 shows that $h$-resolvent is well defined for a strongly convex Legendre function $h$ (thus for our specific $h$ (5)) under the weak monotonicity (implied by the smoothness of $\phi$).

---

> > ### Comment · Reviewer_Mhw9 · 2023-11-23
> >
> > I appreciate the authors for engaging rapidly with the response. I read the updated manuscript more carefully now and I have raised my score to 6. A few remarks:
> >
> > - I think its important to focus on the fact that the paper expands the range of $\rho$ for an explicit scheme. I was admittedly late in noticing that it actually extended the range beyond what is currently known for e.g. EG+.
> > - I understand now that the only Bregman case the paper is concerned with is the choice of $h$ in section 5.1. It would be instructive to compute the Bregman divergence in this particular. How much smaller can the Bregman divergence be in comparison with the square norm even in the best case?
> >
> > I still find the motivation through PGHD for the scheme somewhat vague. In practice we cannot expect to run the implicit scheme, SA, when beyond bilinear problems. So we have to resort to the inexact variant that suffers a logarithmic oracle factor, in which case it is much less clear whether the scheme is favorable from a computational perspective.

---

### Official Review · Reviewer_vygr · 2023-10-31

**Soundness:** 3 good
**Presentation:** 3 good
**Contribution:** 3 good
**Rating:** 6
**Confidence:** 4

**Summary:**

The paper proposed semi-anchored gradient methods to a structured nonconvex-nonconcave minimax problem under certain assumption, namely the weakly Minty variational inequality (MVI). The proposed algorithm is based on the Bregman proximal point (BPP) algorithm, also resembles the primal-dual hybrid gradient (PDHG) method. The proposed algorithm consists of u and v substeps where the authors proposed using FISTA to solve the v substep approximately. Theoretical convergence is studied for this SA-MDGD algorithm and numerical experiments were provided to show the efficacy of the proposed method.

**Strengths:**

The paper is well-rounded and well-motivated. The authors analyzed the theoretical convergence of the general BPP method for a broader class of problems and then proceed to the specific structured problem. The work also addresses the concern of the practicality of the v substep and proposed an inexact SA-MGDA method to carry out the proposed method in practice.

**Weaknesses:**

(Please respond to the questions section directly) It remains unclear how the proposed algorithm performs comparing to the existing works, especially on the theoretical rate of convergence under similar assumptions.

**Questions:**

1. As mentioned in the Weakness section, a comprehensive comparison with GDmax and other algorithms, especially in theoretical convergence rate seems necessary. Is the sublinear rate as in Theorem 4, 5 or 6 show improvements over existing methods or achieve certain lower bounds? For example [1] seems to achieve similar sublinear rate. The authors could consider illustrating this in their Table 1.

2. In numerical experiments, I’m not sure if the authors implemented their SA-GDmax or the more practical SA-MGDA algorithm. If it’s SA-GDmax as in (7), then how did the authors conduct the v substep precisely? Also for Figure 2, the authors claimed the parameter $\tau=0.01$ in section 7.2 but presented two choices of $\tau$, and from the left figure in Figure 2, $\tau=0.01$ didn’t show a statistical advantage of SA-GDmax over other works. Last, the authors didn’t compare with a lot of the methods in Table 1, for which they should consider adding more numerical comparisons.

References:
[1] Diakonikolas, Jelena, Constantinos Daskalakis, and Michael I. Jordan. "Efficient methods for structured nonconvex-nonconcave min-max optimization." International Conference on Artificial Intelligence and Statistics. PMLR, 2021.

---

> ### Author Response · Authors · 2023-11-18
>
> We appreciate the reviewer's nice summary of our paper, and valuable feedback.
> - (Q1) Regarding the clarification of our contribution, see our general comment.
> - (Q2) We implemented the exact SA-GDmax, as we have an efficient max-oracle for both experiments. In particular, the max step of the fair classification problem is exactly maximizing a quadratic function over a simplex, or equivalently, an orthogonal projection problem to a simplex, given the value of the loss function for each category. This can be efficiently done in $O(d_v\log(d_v))$, where $d_v$ is the dimension of $v$ (arXiv: 1309.1541). Note that the orthogonal projection to a simplex is also needed for other methods. / We agree that the experiment with $\tau=0.01$ does not clearly show that our method is better, unlike the case with $\tau=0.001$. We still shared the $\tau=0.01$ result for transparency. / Among extragradient-type methods, we chose CEG+ that is known to work under the weakest condition we consider. We already have ran EG and EG+ and those worked poorly.

---

### Official Review · Reviewer_uoZm · 2023-10-31

**Soundness:** 3 good
**Presentation:** 2 fair
**Contribution:** 2 fair
**Rating:** 6
**Confidence:** 4

**Summary:**

This work for the first time extends the primal-dual hybrid gradient (PDHG) method from convex-concave minimax optimization problem to nonconvex-nonconcave minimax optimization problem. The 4 versions of PDHG (with/without projection and with/without max oracle) obtain the same gradient convergence rate $\mathcal{O}(1/k)$ as the existing extragradient methods, and PDHG without projection and with max oracle upper bounds Bragman distance that is larger than the squared norm measure in the convergence rate of the existing extragradient methods, which yields faster empirical convergence of PDHG without projection and with max over extragradient methods as shown in the experiments.

**Strengths:**

Originality: This work for the first time extends the primal-dual hybrid gradient (PDHG) method from convex-concave minimax optimization problem to nonconvex-nonconcave minimax optimization problem.

Quality: The theoretical and experimental results make sense.

Clarity: Generally I can well understand this paper.

Significance: PDHG without projection and with max oracle upper bounds Bragman distance that is larger than the squared norm measure in the convergence rate of the existing extragradient methods, which yields faster empirical convergence of PDHG without projection and with max over extragradient methods as shown in the experiments.

**Weaknesses:**

The major weakness is the weak advantage of the proposed method over existing works, especially EG+ and CEG+, as elaborated in my questions 1-3 below.

Some typos and unclear points are listed in the questions below.

**Questions:**

(1) In Table 1, is it possible to add some columns to reveal your advantage over EG+ and CEG+? The advantage over EG+ and CEG+ in bounding the larger Bregman distance seems to disappear for SA-GDAmax with projection (Theorem 4) and the inexact SA-MGDA methods (Theorems 5 and 6). Other advantages? Also, I think the practical inexact SA-MGDA methods should also be included in the experiments.

(2) The drawbacks of extragradient and advantages of PDHG could be briefly mentioned in the abstract and the beginning of the Introduction, instead of ''there is still room for improvement''. Also, in the abstract, is the ''worst-case convergence rate'' lacking in extragradient methods? If yes, you could mention this in the abstract.

(3) What's your advantage over the works ''Fast extra gradient methods for smooth structured nonconvex-nonconcave minimax problems'' and ''Stable Nonconvex-Nonconcave Training via Linear Interpolation''? You may cite the latter.

(4) In ''This was studied on a general convex-concave problem, but it has not been found useful in a more general nonconvex-nonconcave minimax problem. In this paper, we demonstrate its natural extension to a structured nonconvex-nonconcave minimax problem'' in the abstract, ''it' and ''its'' are far away from PDHG and thus could be replaced by PDHG.

(5) At the end of the second paragraph of the introduction, "a new nonlinear variant of the PDHG, named semi-anchored (SA) gradient method" could be clearer.

(6) In page 4, in the sentence ''the GDmax minimizes the equivalent minimization problem'', ''minimizes'' could be changed to ''solves''.

(7) How to compute $R(x)$? You could explain or cite in your paper. Can $R(x)$ be exactly solved? If not, it is recommended to include such an error in the convergence results.

(8) In Section 4.3, you said ''This has several advantages over the standard BPP, which will be detailed later. '' Later I found only one advantage of a larger range of $\rho$. Any other advantages?

(9) In Section 5.1,  can we replace $\widehat{L}$ with the previously defined $\gamma$?

(10) Right after as ''it resembles GDmax'', you could indicate that we can also obtain SA-GDmax with projection using BPP with projection (4) using h in (5).

(11) In Theorem 3, ''SA-GDmax (i.e., SA-MGDA with $J=\infty$)'' looks clearer. In Theorem 4, should it be ''SA-GDmax with projection''?

(12) In the toy example, what's the function $\phi$? Should it be $+\frac{L^2\rho}{4}u^2$ and $-\frac{L^2\rho}{4}v^2$ to correspond to $+f(u)$ and $-g(v)$ in the problem (1)?

---

> ### Author Response · Authors · 2023-11-18
>
> We appreciate the reviewer's nice summary of our paper, detailed suggestions and constructive feedback.
> - (Q1-2) Regarding the clarification of our contributions, see our general comment. / As we have an efficient max-oracle for the experiments, the SA-GDmax is not impractical for our experiments and thus SA-MGDA does not seem necessary to be explicitly considered, which behave almost the same as the SA-GDmax.
> - (Q3) They are both interesting, as the former discusses the acceleration, and the latter studies last iterate convergence for the first-time in nonconvex-nonconave problems. However, they both assume the (negative-)comonotonicity that is stronger than the weak MVI that we assume, and how one can extend their results to the weak MVI seem interesting but unclear at the moment. We cited the latter one, next to the former.
> - (Q7) Computing $R(x)$ exactly and inexactly corresponds to SA-GDmax and SA-MGDA, respectively. In particular, the analysis for the inexact computation of $R(x)$ is provided in Theorems 5 and 6 for SA-MGDA (with and without projection). The explicit and detailed analysis of the inexactness of $R(x)$ can be found in Appendix D where we present the proofs of Theorems 5 and 6.
> - (Q8) As mentioned in the general comment, the projection technique was originally introduced to relax the bounded domain assumption in Theorem 5 of the SA-MGDA, which is now removed. So, now there is no advantage other than having a larger range of $\rho$.
> - (Q9) You are right that they can be interchanged in the current form. To clarify this, we moved the definition of $\hat{L}$ earlier in front of Assumption A2, and leave $\gamma$ only for the definition of weak monotonicity.
> - (Q12) The considered function is $\phi$, as written in the paper, and we forgot to inform $f(u)=g(v)=0$, which we added in the revision.
> - (Q4-6,Q10-11) Thanks for the corrections and the suggestions. We have revised accordingly.

---

> > ### Comment · Reviewer_uoZm · 2023-11-19
> > **Clear now. Raise rating.**
> >
> > On one hand, the revised paper is now more clear, especially in the advantage over existing methods such as extragradient.
> > On the other hand, the contribution looks incremental, not very significant.
> > Therefore, I raise my rating to 6: marginal acceptance.
> > Thank the authors for their revision.
> >
> > Reviewer uoZm

---

> > > ### Author Response · Authors · 2023-11-20
> > >
> > > We sincerely appreciate the reviewer for carefully reviewing our rebuttal and expressing a positive view of our work.

---

### Author Response · Authors · 2023-11-18
**General comment on our contributions**

We express our gratitude to all the reviewers for carefully reading our paper and offering valuable feedback. We acknowledge that our contributions were not clearly stated, and we made revisions to the paper to enhance the clarity of our contributions as listed below.

- **We further highlighted why we consider the PDHG-type method in the first place.** In particular, we added a statement, owing from (Chambolle \& Pock, 2011), that the PDHG outperforms the extragradient on a bilinear problem. The SA-GDmax exactly reduces to the PDHG for a bilinear problem, so we expect that the SA-GDmax will outperform the extragradient when entered a locally bilinear problem, regardless of the worst-case rate comparison beyond bilinear setting.
- **We made the theory of the inexact SA-MGDA in Theorem 5 stronger.** First of all, we accidentally missed to mention that the convergence rate of the SA-MGDA in Theorem 5 is in terms of the Bregman distance (and thus in terms of the squared gradient norm), which was already shown in Theorem 7 (of Appendix D.1), a detailed version of Theorem 5. In addition, thanks to reviewer Mhw9, we realized that Theorem 5 does not even need a bounded domain assumption, by simply adopting the proof Theorem 6. (This weakens the necessity of the projection technique in this paper, but it is still useful as it allows larger $\rho$ in the weak MVI condition.)
- **We have made comparison of worst-case rates, under the weak MVI, more apparent.** In particular, thanks to reviewers uoZm and vygr, we modified Table 1 (and the paper), so that our advantage over EG+/CEG+ is more evident; the SA-GDmax and EG+/CEG+ have same $O(1/k)$ rate but the optimality measure (Bregman distance) for SA-GDmax upper bounds that (squared gradient norm) for EG+/CEG+. In addition, we highlighted that, when an efficient max-oracle is available, the SA-GDmax can be superior to EG+/CEG+ even in terms of the computational complexity, which is the case of our numerical experiments. Note that GDmax also has the same $O(1/k)$ rate but only under the PL condition on the max-player, which is not of our interest and does not satisfy for our numerical experiments, so we found mentioning this not necessary.

---

### Meta-Review · Area_Chair_rYor · 2023-12-09

**Metareview:**

The paper studies min-max optimization problems in structured nonconvex-nonconcave primal-dual (PD) settings and proposes new algorithms based on the Bregman proximal point (PP) method. Nonconvex-nonconcave min-max optimization settings are known to be computationally challenging, thus the paper studies them under the "weak MVI" condition proposed in the recent literature. The results in the paper extend the range of the parameter of the weak MVI condition (thus showing that a larger class of nonconvex-nonconcave problems can be addressed algorithmically in poly-time), at the cost of a more expensive oracle access to a problem (Bregman prox for either the entire operator associated with the problem or with the "max" part of the problem). Importantly, it is possible to use an approximate version of the oracle (under additional "smoothness" assumptions for the "max" part function $g$) at a $\log(1/\epsilon)$ cost.

The paper has multiple interesting contributions, but the positioning of the paper seems misplaced. I had trouble understanding this from the author-reviewer discussion, so I read the paper to see what was happening. The main motivations of the paper and its primary "selling points" are: (1) using a PD method in place of extragradient (EG)-type methods, because "they are faster in practice" and (2) introducing a new convergence criterion, which is simply a Bregman divergence between successive iterates. If the paper were to be considered based on these two contributions, then I would agree with the reviewers that this would be incremental and not particularly interesting, for the following reasons. First, the claim about PD methods being "faster in practice" than EG methods is ill-defined and not supported by evidence. The two classes of methods are not directly comparable, because EG applies to a much broader classes of problems (variational inequalities) and assumes only oracle access to evaluations of the operator associated with the problem (plus possibly a projection oracle for the feasible set or its close variant). PD methods apply to more specialized problem classes and usually assume availability of a prox-oracle for the (primal-dual) separable components of the objective. To make a fair comparison, one would need to consider variants of EG specialized to the considered problem class and the available oracle(s). It is known that in terms of the (theoretical) convergence rate, there is no difference between the two classes of methods; one could primarily gain in PD setups from better balancing of the dependence on the problem parameters when the primal and the dual side have substantially different Lipschitz constants and diameters of the feasible set. If EG were to be used in these setups, one would use different step sizes for the primal and the dual portion and that's not hard to handle in its analysis. In any case, if the main motivation is that PD methods are faster in practice, then one would expect to see extensive numerical experiments in place of a single toy experiment in the present paper. Second, Bregman distance between successive iterates is not a surprising measure of convergence for prox-type methods and, more importantly, it is not a good measure of convergence as it is algorithm-specific. One could have a trivial algorithm that always outputs the same point. For such an algorithm, this measure of convergence would be zero (convergence in one iteration!), even though the algorithm is clearly not doing anything useful. Additionally, even the standard analysis of the Mirror-Prox method from Nemirovski's '04 paper bounds an upper bound on the "useful" gap but it is the result for the gap that is stated in the theorems *because it is an interpretable condition.*

Additionally, it should be clarified very early on in the paper that even though the paper considers Bregman PP method, it does not address more general normed spaces than previous work.

That all said, the paper does have good contributions, but it needs to be repositioned. The most interesting ones, from my perspective, come down to extending the range of the weak MVI condition at the cost of a more expensive oracle. Even if the oracle costs $\log(1/\epsilon)$ (or even poly$(1/\epsilon)$), the point is that we get to solve more nonconvex-nonconcave problems (which are computationally challenging in general) than we did before. That should be highlighted better. Second, the paper would benefit from providing clear remarks after the theorem statements explaining by how much we can extend this range by translating its *algorithm* parameters into *problem* parameters. Finally, the result for the quasi-Bregman firm nonexpansiveness of the (Bregman) resolvent under the weak MVI condition appears new to me -- this should be highlighted as well.

There were at least two experts evaluating this paper and understanding its contributions took quite a bit of effort. That begs the question: who is this paper for?

**Justification For Why Not Higher Score:**

I would agree to a higher score, but I think some calibration is needed here. My main reservation is that as currently written, the paper would be useful to a very narrow/niche community.

**Justification For Why Not Lower Score:**

N/A

---

### Decision · Program_Chairs · 2024-01-16

Reject